# Analysis of the Biochemical Effect of Enrofloxacin on American Shad (*Alosa sapidissima*) Infected with *Aeromonas hydrophila*

**DOI:** 10.3390/ani15202962

**Published:** 2025-10-13

**Authors:** Yao Zheng, Jiajia Li, Xiaofei Wang, Kai Chen, Bingwen Xi, Julin Yuan, Gangchun Xu

**Affiliations:** 1Wuxi Fisheries College, Nanjing Agricultural University, Key Laboratory of Freshwater Fisheries and Germplasm Resources Utilization, Ministry of Agriculture and Rural Affairs, Freshwater Fisheries Research Center (FFRC), Chinese Academy of Fishery Sciences (CAFS), Wuxi 214081, China; 2Key Laboratory of Healthy Freshwater Aquaculture, Ministry of Agriculture and Rural Affairs, Key Laboratory of Fish Health and Nutrition of Zhejiang Province, Zhejiang Institute of Freshwater Fisheries, Huzhou 313001, China

**Keywords:** enrofloxacin residue, signaling pathway, metabolome, correlation analysis, healthy culture

## Abstract

**Simple Summary:**

American shad’s culture is increasingly threatened by the potential bacteria and cured by the widespread use of antibiotics, like enrofloxacin (ENR). The present study addresses these issues to analyze the toxic effects of ENR exposure on American shad after 12~48 h, particularly on metabolic changes using transcriptomics and metabolomics. The fluctated enzymatic activities revealed, the pathways including ferroptosis, pyrimidine metabolism, protein processing in endoplasmic reticulum have been significantly enriched, and the key metabolity, L-malic acid showed a highly significant positive correlation with the pathway of pyruvate cycle and citric acid cycle.

**Abstract:**

In order to find the biochemical effects of *Aeromonas hydrophila* and its therapeutic chemical, enrofloxacin (ENR), on American shad (*Alosa sapidissima* A. Wilson), four groups were set up: a control group (C), an *A. hydrophila* group (A), an *A. hydrophila* + 70 mg·L^−1^ enrofloxacin (ENR) group (E1), and an *A. hydrophila* + 140 mg·L^−1^ ENR group (E2). Histological, enzymatic activities, transcriptome, and proteomics have been performed. MDA, PPO, AKP, TNF-α, and AMPK were significantly increased, while AhR and EROD were decreased in the liver of American shad after treatment with *A. hydrophila*. AhR and EROD showed a significant decrease in E1 group; MDA, PPO, AKP, and AMPK were significantly increased, while AhR and EROD decreased in E2 group. *A. hydrophila* significantly increased ferroptosis, TGF-β signaling pathway, etc. Ferroptosis, pyrimidine metabolism, and glycerolipid metabolism significantly increased in E1 group, while protein processing in endoplasmic reticulum significantly increased in E2 group. A total of 126 shared metabolites were found in the comparisons of A vs. C and E2 vs. C, and the main enriched pathway were organic oxygen compounds, lipids, and lipid-like molecules. Except for fluorobenzoate degradation, the pathways of ascorbate and aldarate metabolism, pyrimidine metabolism significantly increased in A and E2 groups, which further resulted in vacuolization, cell shedding, and necrosis in the liver. *A. hydrophila* led to a significant decrease in lipid metabolism, leading to oxidative stress and energy expenditure. The addition of ENR in aquaculture significantly enhanced liver metabolic abnormalities caused by *A. hydrophila*. Excessive use of ENR leads to oxidative stress in American shad, affecting its immune system as well as lipid, carbohydrate, and energy metabolism.

## 1. Introduction

The American shad (*Alosa sapidissima*), originally introduced from the United States [1], has become one of the most valuable aquaculture species in China. In premium restaurants, it can sell for as much as USD 80–100 per kilogram, making it a luxury product in the domestic market. This high economic value has spurred a wide range of research, including studies on reproductive biology [2,3,4], nutritional composition, disease protection [5,6], culture mode, and even its applications. Despite these advances, farming American shad remains difficult. Outbreaks of bacterial and parasitic diseases are frequent, and the species is highly sensitive to changes in its environment [6]. These challenges are particularly acute in recirculating aquaculture systems (RAS) and reservoir cage farming, where shad are commonly raised.

Fish deaths in the wild, aquaculture, and ponds are often linked to bacterial diseases, with *Aeromonas hydrophila* being the most common freshwater pathogen in China, especially in American shad culture [5]. This Gram-negative, non-spore-forming bacterium belongs to the genus *Aeromonas* (family Vibrionaceae) and thrives in soil, sewage, and river sediments. It infects a wide range of hosts, including fish, crustaceans, amphibians, livestock, and even humans. In fish, especially cyprinids in China, *A. hydrophila* causes epizootic septicemia, leading to hemorrhagic disease and high mortality. The bacterium typically enters through the gut, spreads systemically, and its virulence is strongly influenced by environmental stress, such as poor water quality or co-infection with other pathogens. Infection triggers immune responses, oxidative stress, and liver fat accumulation. Frequent outbreaks result in heavy economic losses and remain a major challenge to sustainable aquaculture [4,5].

To combat bacterial infections, antibiotics such as enrofloxacin (ENR) are widely applied in Chinese aquaculture [7]. However, frequent and often excessive use has resulted in the persistence of ENR in surrounding waters (3.49~660.13 ng·L^−1^), sediments (1.03~722.18 μg·kg^−1^), and even fish tissues (6.73~968.66 μg·kg^−1^). Long-term ENR exposure has been associated with oxidative stress, metabolic disruption, and inflammation in cultured species, including American shad [7,8]. Monitoring studies across China have confirmed the widespread occurrence of ENR in aquaculture environments [7]. In Lake Honghu of Hubei Province and ponds in Guangxi Province, for instance, ENR was detected in water, sediments, and fish tissues throughout the year, sometimes reaching several micrograms per kilogram [9,10]. Higher ENR levels were reported in fish ponds compared with shrimp or crab ponds in Lake Taihu [11]. In Guangdong Province, ENR residues in pond water and sediments reached 21.3 ng·L^−1^ and 446 μg·kg^−1^, respectively [12], while in the Laizhou Bay marine farms, residues were also found in both water and sediment [13]. These compounds not only persisted in the environment but also accumulated in fish tissues, raising concerns about bioaccumulation, food safety, and the spread of antibiotic resistance genes (ARGs) [13,14]. The ecotoxicological risks of ENR were significant because ENR residues were widely present during both use and non-use periods. The predicted no-effect concentration for algae was only 4.9 μg·L^−1^ [15], and in tilapia, ENR displayed a long elimination half-life (21.7 h) and low clearance rate (0.09 L·h^−1^·kg^−1^), suggesting strong persistence in aquatic organisms [16]. For American shad specifically, ENR exposure was linked to gut microbiota disruption, oxidative stress, and altered metabolic pathways that influence growth and body weight [7,8].

The accumulation of ENR in water and sediment poses potential harm to fish, and gene expression studies have revealed the interaction between T helper cells and macrophages in the fish immune system’s response to *A. hydrophila* and infection memory. Transcriptomics and metabolomics play an important role in identifying key signaling pathways and gene regulatory networks involved in the treatment process of antibiotic induced *A. hydrophila* infection and contribute to a better understanding of the complex processes of antibiotic action on the fish body. Water quality plays a central role in the success of fish farming. In intensive aquaculture systems, hydrodynamics further complicate management by altering the pharmacokinetics of administered drugs. For example, flowing water has been shown to accelerate the clearance of norfloxacin from fish tissues, while simultaneously promoting its accumulation [17]. This means that different concentrations of ENR have varying effects on fish itself. In recent years, multi-omics approaches have been employed to elucidate the molecular mechanisms of toxic stress in tilapia [18,19] and American shad [20]. Advances in omics technologies have provided powerful tools to uncover biological responses at multiple levels. Transcriptomics have enabled accurate detection and quantification of gene expression, offering insights into molecular pathways affected by stress. Metabolomics complements this by profiling organism-wide metabolite changes under environmental challenges. Integrating transcriptomics and metabolomics has enhanced our understanding of the molecular and metabolic mechanisms underlying fish responses to contaminants. For microalgae, a test concentration of 50 mg·L^−1^ ENR has been chosen for toxicological evaluation. Although ENR is widely used, frequent or excessive application can leave antibiotic residues in the environment, and longer exposures are likely to produce more severe effects. The acute toxicity of ENR, however, has not yet been clearly established.

Given these challenges, there was a clear need for a dual approach: the key affecting pathways and antibiotic ENR residues in American shad aquaculture. The present study addressed these issues to analyze the toxic effects of acute ENR exposure on American shad for 12~48 h based on the LC_50_ value, particularly on metabolic changes using transcriptomics and metabolomics.

## 2. Materials and Methods

### 2.1. Experimental Setup and Sampling

Juvenile American shad *Alosa sapidissima* (*n* = 180, 2.79 ± 0.84 g, 3.51 ± 0.25 cm, for the LC_50_ test) and (*n* = 240, 86.11 ± 8.26 g, 18.99 ± 0.81 cm) were obtained from Yangzhong Jiang Zhiyuan, Food Industry Co., Ltd. (119°82′ E, 32°30′ N), and the exposure fish were randomly selected. Fish were maintained in polypropylene plastic tanks (with diameter 2 m, height 1.5 m). About 200 L of water were injected into the cylinder, and on a daily basis, about one-third of the water was changed. Continuous oxygenation was kept up during the experiment, and an air conditioner was used to regulate the temperature of the breeding environment. The control water temperature was 17 ± 1 °C, dissolved oxygen was 6.9 ± 0.2 mg·L^−1^, pH was 6.9 ± 0.1, and nitrogen and phosphorus contents met the fishery water quality standard. Fish were fed twice a day at 8:00 a.m. and 16:00 p.m. The daily feeding amount was based on the technical specification for *Alosa sapidissima*, which is 4% of the body weight of the experimental fish. Animal welfare was prioritized in accordance with FFRC-CAFS regulations (LAECFFRC-2023-07-14).

For the 48 h LC_50_ test for finding the range of ENR addition dosage, a static water contact acute test method was used to determine the 48 h LC_50_ of ENR on juvenile American shad. The upper and lower limits of ENR dosage were 750 and 550 mg·L^−1^, respectively. The concentration gradient of ENR was set using the arithmetic sequence method, with 3 replicates per concentration gradient with 6 groups and 10 fish per replicate (*n* = 180). The six mass concentration gradients were 0 mg·L^−1^, 550 mg·L^−1^, 600 mg·L^−1^, 650 mg·L^−1^, 700 mg·L^−1^, and 750 mg·L^−1^, respectively, and the aquaculture water was 10 L. Once the experiment began, the fish were monitored continuously for the first 6 h, after which their condition was recorded every 6 h, and any dead individuals were promptly removed. The LC_50_ was calculated using Excel based on the drug concentration and mortality rate (calculated for 6 h, 12 h, 24 h, 48 h). According to the Tumbell formula, the safe mass concentration was calculated as SC = 48 h LC_50_ × 0.3/(24 h LC_50_/48 h LC_50_).

The strain of *Aeromonas hydrophila* (SHBCC D11385) used in this experiment was obtained from the Shanghai Bioresource Collection Center (Shanghai, China), following a previously published report [21]. The bacteria were first cultured in LB medium at 28 °C for 24 h. After incubation, the culture was transferred to fresh medium for further growth and differentiation, and the bacterial concentration was determined using a UV spectrophotometer at 600 nm. The culture was then centrifuged at 12,000 g for 15 min, the supernatant was discarded, and the pellet was resuspended in 0.65% sterile normal saline. The suspension was adjusted to a final concentration of 1 × 10^6^ CFU·mL^−1^. Prior to injection, the fish were lightly anesthetized with MS-222 (10 mg·L^−1^ for 15 min). At the start of the experiment, the fish were intraperitoneally injected with a single dose of *A. hydrophila* (LD_50_ = 1.0 × 10^6^ CFU·mL^−1^), after which they were returned to the breeding tanks.

In the static exposure trial, 240 fish were divided into four treatment groups, with each group replicated across three tanks (20 fish per tank). The groups were as follows: a control group without *A. hydrophila* or ENR (C); a group infected with *A. hydrophila* (1.0 × 10^6^ CFU·mL^−1^, A); a group infected with *A. hydrophila* and treated with 70 mg·L^−1^ ENR (E1, the standard dose used in American shad culture); and a group infected with *A. hydrophila* and treated with 140 mg·L^−1^ ENR (E2, double the standard dose, here considered excessive). Fish were exposed for 12, 24, and 48 h. The ENR concentrations were chosen based on earlier studies that applied 50–100 mg·L^−1^ ENR to microalgae [22,23]. Each treatment was conducted in triplicate using separate glass tanks. Sampling took place at 12, 24, and 48 h, and groups were labeled according to treatment and sampling time (e.g., CL12, AL12, E1L12, E2L12 for 12 h; CL24, AL24, E1L24, E2L24 for 24 h; CL48, AL48, E1L48, E2L48 for 48 h). Here, “L” indicates liver samples. From each tank, six fish were collected at the designated time points. Liver samples were processed for hematoxylin and eosin (H&E, Nanjing Jiancheng Bioengineering Research Institute, Nanjing, China) staining (*n* = 3, at 12–48 h), RNA-seq (*n* = 3, at 48 h), qPCR (*n* = 3, at 48 h), and metabolomic analysis (*n* = 5, including three fish overlapping with other assays and two additional fish from the same tank at 48 h). For metabolomic assays, “S” denotes the corresponding samples. Prior to all procedures, fish were euthanized with MS-222 to minimize stress and ensure humane handling.

### 2.2. Hepatic Histopathological Alterations, ENR Determination in Muscle, Hepatic Enzymatic Activities

Liver samples (*n* = 3 per tank) were fixed in 4% formaldehyde for 24 h, then processed through routine washing, graded dehydration, clearing, wax infiltration, and embedding. Tissue sections were cut into 5 μm slices using a rotary microtome (Leica RM2235, Leica Microsystems, Vizna, Germany). These sections were subsequently dewaxed, rehydrated, stained with hematoxylin and eosin (H&E), dried, sealed with neutral gum, and examined under a light microscope at 400× magnification (Olympus CHC binocular, Tokyo, Japan).

To measure ENR residues in American shad, 2.00 ± 0.02 g of muscle tissue was accurately weighed into a 50 mL centrifuge tube. Ten milliliters of extraction solution (0.1% formic acid in acetonitrile) were added, and the mixture was vortexed at 2000 r·min^−1^ for 15 min before centrifugation at 12,000 r·min^−1^ for 5 min. Solid-phase extraction cartridges were first activated with 5 mL methanol and equilibrated with 5 mL water. Two milliliters of the supernatant were then loaded onto the cartridge and eluted twice with formic acid methanol solution (3 mL and 2 mL). The eluate was collected in a 200 mL round-bottom flask, evaporated to dryness in a 45 °C water bath under reduced pressure (0.07 MPa), reconstituted in 1 mL of the initial mobile phase, and filtered through a 0.22 μm organic phase membrane. Chromatographic analysis was performed on an Agilent ZORBAX SB-C18 column (150 × 4.6 mm, 5 µm, Santa Clara, CA, USA). The mobile phase consisted of acetonitrile and 0.01 mol·L^−1^ tetrabutylammonium bromide (pH adjusted to 2.8 with phosphoric acid) at a ratio of 10:90 (*v*/*v*), pre-filtered and degassed. The column was maintained at 30 °C with a flow rate of 1.0 mL·min^−1^. Detection was carried out using a fluorescence detector (excitation 280 nm, emission 450 nm), with a 10 µL injection volume.

The liver tissue was homogenized with PBS at a ratio of 1:9, whereafter the supernatant was collected and stored in a −20 °C freezer until analysis was performed. Firstly, protein quantification was performed using the Coomassie Brilliant Blue method. The enzymatic biomarkers were bought from the Nanjing Jiancheng Bioengineering Institute (Nanjing, China) containing total protein (TP, g·L^−1^), alkaline phosphatase (AKP, U·gprot^−1^), acid phosphatase (ACP, U·gprot^−1^), polyphenol oxidase enzyme (PPO, U·mgprot ^−1^), trace malondialdehyde (MDA, nmol·mgprot ^−1^), acquired cytochrome p450 1a1 (EROD, from Jiangsu Meimian Industrial Co., Ltd., pg·mL^−1^, Yancheng, China), Adenosine 5‘-monophosphate (AMP)-activated protein kinase (AMPK, µg·L^−1^), tumor necrosis factor α (TNF-α, µg·L^−1^), aryl hydrocarbon receptor (AhR, pg·mL^−1^), and ryanodine receptor (RyR, pg·mL^−1^), and data was analyzed according to our previous studies [18,19,20] after being measured using a Jasco-V530 spectrophotometer (Beijing, China).

### 2.3. RNA-Seq

RNA integrity (samples chosen RNA integrity value ≥ 7) and concentration were determined using an RNA Kit (American life technology company, Seattle, USA). A total of 1 μg RNA was used for each liver sample as input material for RNA-Seq sample preparation, the PCR product (ampurexp system) was purified, and the library quality was evaluated using the Agilent Bioanalyzer 2100 system. To ensure the quality of the transcriptome analysis, filtering of the original sequencing data (raw reads), DGEs enrichment, and KEGG pathway analysis (KOBAS) were performed.

To further identify the ENR treatment-related genes as target genes, we screened the significant DEGs in the KEGG pathways associated with ferroptosis, protein processing in endoplasmic reticulum, folate biosynthesis, glycine, serine, and threonine metabolism for verification (q-PCR). The efficiency (*E*) of each PCR was determined by the slope generated using a 10-fold diluted cDNA series with five dilution points in triplicate. The equation used was *E* = 10 ^(−1/slope)^. *β-actin* was chosen as the reference gene because its expression remained constant among the experimental groups, and the cDNAs for the gene expression analysis were normalized to *β-actin* (Table A1). Changes in the mRNA levels of these genes were calculated according to a previously described method [18,19,20].

### 2.4. Metabolome and Combined Analysis

We weighed 50 mg of the sample and added it to 1000 μL of an internal standard extraction solution (which is a mixture of methanol, acetonitrile, and water in a volume ratio of 2:2:1, with an internal standard concentration of 20 mg·L^−1^). After vortexing, grinding, sonicating, and centrifuging, we took 500 μL of supernatant and vacuum-dried it. We added 160 μL of acetonitrile water extraction solution (volume ratio 1:1) to the dried sample and dissolved it again. After vortexing, sonicating, and centrifuging, we took 120 μL of supernatant and injected it into a 2 mL vial. We took 10 μL of each sample and mixed it to form a quality control sample for subsequent instrument testing.

Five liver sample supernatants per group were obtained from American shad samples and analyzed using LC-MS following the reference [18,20]. We imported the raw data into Mass Profiler software for preprocessing. After processing, the data was imported into Excel 2020 and converted into a two-dimensional data matrix. This matrix provides detailed records of retention time, mass to charge ratio, sample information, peak intensity, and other related data. We extracted multiple different samples in equal quantities and mixed them to make control (C) samples. During the testing process, control samples were added to the test sequence at regular time intervals to evaluate the stability and reliability of the analytical equipment by comparing the consistency of chromatograms. After data processing, we imported it into SIMCA-P 13.0 software. We normalized the data before analysis and used OPLS-DA statistical analysis to compare the data of each group, and different expression metabolites (DEMs) were computed and analyzed with the selection thresholds (|FC| > 1, *p*-value < 0.05, VIP > 1). The KEGG database was used to analyze the metabolic pathways corresponding to these differential metabolites, and bubble plots were used to represent metabolic pathway enrichment analysis. The conditions for electrospray ionization mode contained the detection mode (multi-reaction), the ion source and desorption temperatures (150 °C and 550 °C), and the dissolved gas flow (1000 L·Hr^−1^). Genes, enzymatic activities, and metabolites related to the above pathways were analyzed to identify signaling molecules participating in the same pathway at 48 h.

### 2.5. Data Statistical Analysis

All experimental data were processed using SPSS 26.0 software and presented as mean ± standard deviation (SD). For statistical testing, data that did not conform to normal distribution or homogeneity of variance were log_2_-transformed to meet the assumptions of parametric analysis.

One-way analysis of variance (ANOVA) was employed to assess differences among treatment groups. When significant variation was detected (*p* < 0.05), Tukey–Kramer post hoc tests were conducted to determine specific group differences. A significance level of *p* < 0.05 was considered statistically meaningful throughout all comparisons.

## 3. Results

### 3.1. LC_50_ for ENR on Juvenile American Shad

As the concentration of ENR increased, the mortality rate of juvenile American shad increased. The mortality rates at 24 h were 26.67%, 43.33%, 63.33%, 76.67%, and 93.33%, respectively (Figure 1). The mortality rates at 48 h were 33.33%, 53.33%, 76.67%, 96.67%, and 100%, respectively. The LC_50_ of ENR for juvenile American shad at 24 and 48 h were 624.21 mg·L^−1^ and 593.37 mg·L^−1^, respectively, with a safe mass concentration of 169.22 mg·L^−1^, which referred to the exposure concentration selection for further studies.

### 3.2. Histopathological Alterations Induced by A. hydrophila and Normal and Excessive Concentrations of the Therapeutic Drug ENR

In the control groups, the liver cells maintained a normal appearance, showing uniform distribution, regular shapes, distinct nuclear staining, and clear boundaries (Figure 2). In group A, liver changes began to appear at 12–24 h, including cell enlargement, atrophy, and necrosis, with vacuolization becoming more evident by 48 h. In the E1 group, slight hepatocyte atrophy was noted at 48 h. The most severe changes were seen in the E2 group at 48 h, where the liver tissue showed darker staining, marked deformation, loss of clear cell structure, widespread atrophy and vacuolization, along with extensive cell shedding and necrosis.

### 3.3. ENR Residues in Shad Muscle at 12~48 h

ENR were detected in the fish muscle, with a detection rate of 100%. ENR (10.67~15.33 mg·kg^−1^ in E1, 13.24~17.14 mg·kg^−1^ in E2) in ENR treatment groups was higher than those in other sample points (control and *A. hydrophila* affecting groups), and the average ENR residue concentrations were in the following order: E2 > E1 > C >A (Figure 3).

### 3.4. Hepatic Enzymatic Activities at 48 h

To explore the mode of action under ENR exposure, hepatic enzymatic activities, transcriptome, metabolome, and their integrated analysis were performed at 48 h (Figure 4). The MDA content was significantly elevated in groups A and E2 compared to groups C and E1, with group E2 reaching the highest level of 211.97 nmol·mgprot^−1^. Similarly, PPO activity increased in groups A and E2 relative to the control (*p* < 0.05), while AKP activity was significantly higher in groups E2 and A than in C and E1, peaking at 52.54 U·gprot^−1^ in E2. AMPK activity also showed a marked increase in groups E2 and A compared with C and E1 (*p* < 0.05), whereas PPO activity in E1 was lower than in A.

ACP activity decreased in group A compared to E1 (*p* < 0.05), while TNF-α levels in A were higher than in all other groups (*p* < 0.05). EROD and AhR contents were significantly reduced in groups A, E1, and E2 (*p* < 0.05). Notably, ACP activity in E1 was slightly higher than in C, while A showed lower ACP compared to C and E2. AMPK activity in E1 was higher than C but not significantly (*p* > 0.05). PPO and MDA levels in E1 did not differ significantly from C or E2 (*p* > 0.05), and although AKP in E1 was higher than C, this was not statistically significant. TNF-α in E1 and E2 remained elevated relative to C, while RyR levels were slightly lower in all treated groups compared to the control, but these changes were not significant (*p* > 0.05).

### 3.5. RNA-Seq After ENR Exposure

After sequencing quality control, a total of 6.14 Gb clean data was obtained, and the Q30 base percentage of each sample was not less than 94.80%. A total of 69,373 single genes with an average length of 1476 bp were obtained, and approximately 21,678 DEGs were detected in each group. In addition, 23,478 annotated genes showed no significant difference in this study. A total of 5969 DEGs were identified in the control group, including 2924 up-regulated genes and 3045 down regulated genes. Comparative analysis showed that 1890 individual genes had significant expression differences, including 256, 175, and 628 genes upregulated within 48 h after infection, as well as 303, 170, and 809 genes downregulated in the comparisons CL48 vs. AL48, CL48 vs. E1L48, and CL48 vs. E2L48 (Figure 5a).

A total of 231 DEGs annotated in CL48 vs. AL48 were enriched in 124 pathways, with significant enrichment (*p* < 0.05) in the pathways ferroptosis, TGF-β signaling pathway, apelin signaling pathway, arachidonic acid metabolism, and vascular smooth muscle contraction. In the comparison of CL48 vs. E1L48 (Figure 5b), 155 DEGs annotated in the middle were enriched in 120 pathways, with significant enrichment observed ferroptosis, pyrimidine metabolism, glycerolipid metabolism, pantothenate, and CoA biosynthesis. The 683 DEGs annotated in CL48 vs. E2L48 were enriched in 188 pathways, among which 50 pathways were significantly enriched (*p* < 0.05), mainly related to the metabolism and cellular processes of the three major nutrients. The CL48 vs. E1L48 comparison group and the CL48 vs. E2L48 comparison group had a total of 111 pathways, among which two pathways were significantly enriched (*p* < 0.05), namely folate biosynthesis, glycine, serine, and threonine metabolism.

Enrichment and pathway analysis were conducted on DEGs to understand the functions of immune related genes. Our research results indicated that many important functional genes related to complement and coagulation cascade, chemokine signaling proteins, and immunity are expressed. Twenty-four selected shared DEGs were validated using qPCR (Table 1). In the CL48 vs. E2L48 group, the *loc121684913* and *sptb* genes were upregulated compared to the other two comparison groups, while the *loc121700164*, *chka*, *igfbp1a*, *klf10*, *sid_key-34d22.1*, *slc10a4*, and *slc3a2b* genes were downregulated compared to the other two comparison groups.

### 3.6. Metabolome

Five samples were used for metabolome analysis, which showed good sample repeatability (Figure 6a). In the comparison for the C (control) group and the A (*A. hydrophila*) group, there were 320 differential metabolites (DEMs) (243 up-regulated and 77 down regulated), and in the E2 vs. C comparison group, there were 332 DEMs (217 up-regulated and 115 down regulated). To screen for common or unique DEMs, there were 194 unique DEMs between CS48 (“S” stands for metabolome samples) and AS48, and 206 unique DEMs between CS48 and E2S48. There were 126 common DEMs among the three treatment groups (Figure 6b). After KEGG annotation, 265 DEMs were annotated, including 115 positive ion DEMs and 150 negative ion DEMs. Based on the positive ion POS mode, 161 DEMs were enriched in 82 KEGG pathways in CS48 vs. AS48, and 113 DEMs were enriched in 53 KEGG pathways in CS48 vs. E2S48.

Among the top 20 pathways significantly enriched in CS48 vs. AS48, fluorobenzoate degradation, pentose phosphate pathway, inositol phosphate metabolism, fructose and mannose metabolism, pyrimidine metabolism, benzoate degradation, purine metabolism, glycolysis/gluconeogenesis, AMPK signaling pathway, phenylalanine, tyrosine and tryptophan biosynthesis, phosphonate and phosphinate metabolism, and acarbose and validamycin biosynthesis significantly increased (Figure 6c). Among the top 20 pathways significantly enriched in CS48 vs. E2S48, fluorobenzoate degradation (Figure 6d), chlorocyclohexane and chlorobenzene degradation, fructose and mannose metabolism, ascorbate and aldarate metabolism, glucagon signaling pathway, benzoate degradation, insulin resistance, diabetic cardiomyopathy, biosynthesis of phenylpropanoids, biosynthesis of alkaloids derived from histidine and purine, renal cell carcinoma, phenylalanine, tyrosine and tryptophan biosynthesis, acarbose and validamycin biosynthesis, and glyoxylate and dicarboxylate metabolism increased.

### 3.7. The Combined Analysis Under ENR Exposure

Comparing the pathways jointly involved by DEGs in the transcriptome and DEMs in the metabolome, there were 41 pathways jointly involved in CS48 vs. AS48 and six pathways significantly enriched in differentially expressed genes or metabolites (*p* < 0.05), respectively. It included ferroptosis, ascorbate and aldarate metabolism, arachidonic acid metabolism, inositol phosphate metabolism, pentose phosphate pathway, and pentose and glucuronate interconversions. With the comparison of CS48 vs. E2S48, there were 66 co-participating pathways, among which fructose and mannose metabolism, carbon metabolism, ascorbate and aldarate metabolism, glyoxylate and dicarboxylate metabolism, pentose and glucuronate interconversions, citrate TCA cycle, pyruvate metabolism, and amino sugar and nucleotide sugar metabolism were notable. The differentially expressed genes and metabolites were significantly enriched (*p* < 0.05), most of which were related to energy metabolism (Table 2).

After drawing a correlation clustering heatmap of DEGs and DEMs with a correlation coefficient greater than 0.8 in the selected pathway, results showed that group C and group A (Figure 7a) showed a significant negative correlation (*p* < 0.05) between alpha-D-Ribose 1-phosphate in the pentose phosphate pathway and *loc121697229*. D-Galactino-1,5-lactone in ascorbic acid and aldehyde metabolism was significantly negatively correlated with *loc121715106* (*p* < 0.05), while L-xylo-Hexulolactone was significantly positively correlated with *miox* (*p* < 0.05). UDP glucose was significantly positively correlated with *agla* in starch and sucrose metabolism (*p* < 0.01) and *gbe1b* (*p* < 0.05). Melibiose was significantly positively correlated with *abca5* in ABC transporter protein (*p* < 0.05).

Eight significantly enriched pathways were further analyzed based on the selected DEGs and DEMs (*p* < 0.05) from the comparison between group C and group E2 (Figure 7b), including DEMs in the pathway of fructose and mannose metabolism, like beta-D-Fructose 2,6-bisphosphate, D-Fructose 1-phosphate, Glyceraldehyde 3-phosphate, and L-Fucose 1-phosphate. 5-Formyl-5,6,7,8-tetrahydroethanopterin in the carbon metabolism pathway was significantly positively correlated with *tpi1b*, *echs1*, *pfklb*, *amt*, and *glud1b* (*p* < 0.01) and significantly negatively correlated with *loc121705140* (*p* < 0.01). Glyceraldehyde-3-phosphate was significantly positively correlated with *cat* and *idh2* (*p* < 0.01), significantly positively correlated with *idh2* (*p* < 0.01), and significantly negatively correlated with *loc121688100* (*p* < 0.01). In the metabolism of ascorbic acid and aldehyde, (4R, 5S) -4,5,6-trihydroxy-2,3-dioxohexanoate was significantly positively correlated with *ugt5d1* and *loc121681730* (*p* < 0.01). D-Galactarate was significantly positively correlated with *loc121690457* (*p* < 0.01), and D-Galactino-1,5-lactone was significantly positively correlated with *loc121690457* and *loc121681730* (*p* < 0.01). There was a significant positive correlation (*p* < 0.01) between (4R, 5S)-4,5,6-trihydroxy-2,3-dioxohexanoate and *ugt5d1*, *zgc_56622*, *loc121681730* in pentose and glucose metabolism and a significant positive correlation (*p* < 0.01) between D-Xylnolactone and *loc121681730*; L-malic acid in the citric acid cycle (TCA cycle) was significantly positively correlated with *pck2* (*p* < 0.001), significantly positively correlated with *idh2* (*p* < 0.01), and significantly negatively correlated with *loc121688100* (*p* < 0.01).

In the metabolism of amino and nucleoside sugars, CMP-pseudaminic acid, L-Fuchose 1-phosphate was significantly negatively correlated with *gmppb* and *gales* (*p* < 0.01), while GDP-4-acetamido-4,6-dideoxy-alpha-D-mannose was significantly negatively correlated with *gnpnat1* and *pmm2* (*p* < 0.01). After screening, the differential genes (*pck2*) and metabolites (L-malic acid) in the pyruvate cycle (Figure 8a) and citric acid cycle (TCA cycle, Figure 8b) showed a highly significant positive correlation (*p* < 0.001). The upregulation of *pck2* gene expression and the increase in L-malic acid reflected the body’s adaptive adjustment to high concentration ENR environment to maintain the energy supply (Figure 8c) and metabolic balance of liver cells.

## 4. Discussion

### 4.1. Environmental Antibiotics in American Shad Culture

ENR is one of the commonly used antibiotics, and its concentration in aquatic environments in China has reached 5681.90 ng·L^−1^. The 24 h and 48 h LC_50_ of ENR for the juvenile German framed carp were 693 mg·L^−1^ and 629 mg·L^−1^, with a safe quality concentration of 155 mg·L^−1^. The LC_50_ of ENR for carp larvae at 24 h and 48 h were 746.75 mg·L^−1^ and 713.61 mg·L^−1^, respectively, and the safe mass concentration was 204.58 mg·L^−1^, which was higher than those in juvenile American shad (safe mass concentration of 169.22 mg·L^−1^, and the exposure concentrations were 70 and 140 mg·L^−1^). In the normal ENR (group E1), the detected enzymatic activities, except for EROD and AhR, showed no significant difference when compared with controls. Both exposure to *A. hydrophila* and excessive dosage of therapeutic chemical ENR can lead to an increase in antioxidant activity and AKP activity in the liver of American shad. ENR can inhibit the activity of EROD, AhR, and inflammatory factors in the liver of American shad. AKP, ACP, AMPK, and TNF-α all showed an upward trend under exposure to *A. hydrophila* and 140 mg·L^−1^ ENR. Within this context, the widespread detection of ENR and its metabolite ciprofloxacin (CIP) in both water and sediment is concerning, which was confirmed in this study, even in the control fish’s muscle, which may be relative to environmental contamination in the ponds of China. These factors explain the persistence of ENR residues across seasons and highlight the need for targeted remediation strategies, which found 13.24~17.14 mg·kg^−1^ ENR in E2 group with detection rate as 100%, even in the control group without ENR. ENR is known to disturb lipid metabolism, provoke oxidative damage, and trigger inflammatory pathways in fish [7,8,14]. Because of its strong affinity for sediments with high organic content [24], ENR tends to accumulate in pond bottoms, particularly in cement-lined systems where microbial diversity is low and degradation capacity is poor. One of the more difficult challenges is the persistence of ENR in sediments. The drug’s elimination half-life in fish contributes 20 h [25], which contributed to chronic liver, intestinal, and oxidative stress injuries [7,8]. Further studies focusing on the harm on ENR to human health through food chain need to be performed.

### 4.2. Mode of Action After ENR Exposure

American shad has emerged as a promising species for recirculating aquaculture systems (RAS) in China [7,26]. Genomic and transcriptomic studies [27,28,29] have investigated the effects of elevated temperatures on American shad [29]. Elevated temperatures increase metabolic demand and physiological stress in shad, though these effects can be partially offset by dietary supplements such as probiotics. For instance, *Lactococcus lactis* has been shown to stimulate fatty acid β-oxidation, reduce oxidative stress, and lower cortisol secretion to counteract a higher toxicity when compared with ENR’s parent compound [30,31,32]. In this study of transcriptome, ferroptosis, TGF-β signaling pathway, pyrimidine metabolism (which was demonstrated in shad from the reference [30]), and protein processing in endoplasmic reticulum were significantly enriched, while in the metabolome’s result, biosynthesis of plant secondary metabolites [22], fluorobenzoate degradation, pyrimidine metabolism [30], ascorbate and aldarate metabolism, chlorocyclohexane and chlorobenzene degradation, and fructose and mannose metabolism [33] were significantly enriched. The results of joint omics analysis showed that exposure to high concentrations of ENR resulted in changes in carbohydrate metabolism, lipid metabolism, cellular processes, and immune system pathways in the liver of American shad.

After the combined analysis, *pck2* and L-malic acid showed a highly significant positive correlation with the pathway of pyruvate cycle and citric acid cycle. L-malic acid is an important intermediate in the TCA cycle, which can be converted to oxaloacetate (a substrate of phosphoenolpyruvate carboxykinase) through the action of malate dehydrogenase. The elevated activities of pyruvate kinase lead to mitochondrial function and energy metabolism imbalance found in marine medaka embryos exposed to 100 μg·L^−1^ semicarbazide for 14 d [34]. The pyruvate cycle has been shown to play a critical role in mediating aminoglycoside antibiotic killing [35] and enhancing membrane permeability [36], which may be helpful for antibiotic susceptibility/resistance of bacteria. However, the citric acid cycle pathway has been identified as a potential key modulator of antibiotic resistance in poultry *Gallibacterium anatis* [37]. The citric acid cycle pathway is one of the main metabolic pathways in cells, responsible for converting acetyl CoA into carbon dioxide and water, while producing ATP as an energy source. The *pck2* gene promotes the flow of TCA cycle intermediates (such as oxaloacetate) to gluconeogenesis pathways, thereby maintaining cellular energy and metabolic balance. After excessive dosage of ENR toxicity, the liver of American shad maintained stability through inflammatory immune response, antioxidant stress, increased energy metabolism, and activation of immune system related pathways.

Biological degradation of ENR, whether by bacteria [38] or algae, is increasingly viewed as a practical and sustainable approach for aquaculture water treatment. At the same time, pond hydrodynamics—whether water is flowing or stagnant—plays a decisive role in shaping antibiotic distribution and removal efficiency [17]. Although thermal hydrolysis at 160–170 °C can remove more than 80% of ENR, such harsh conditions are impractical for aquaculture [39]. To address this, integrated remediation approaches are gaining attention, including bacterial–algal consortia and biochar-based carriers that can stabilize microbes while enhancing adsorption capacity [40]. These strategies offer a way forward for tackling residues embedded in sediments, not just in the water column. Antibiotics in aquaculture rarely occur in isolation. They co-exist with heavy metals, organic pollutants, and sometimes microplastics [41,42], creating a cocktail of stressors that compromise fish health and increase risks to consumers through trophic transfer [18]. Of course, several limitations remain. We were not able to determine the full mode of action using orthogonal design, and the precise pathways of ENR catabolism remain unresolved. By linking molecular insights with ecological engineering, it is possible to reduce antibiotic contamination, safeguard the welfare of cultured species such as American shad, and mitigate risks to both environmental and human health.

## 5. Conclusions

We showed that, after ENR exposure, several liver enzymes like MDA, PPO, AKP, and AMPK significantly increased, whereas AhR and EROD significantly decreased in American shad. From the results of transcriptome, ferroptosis, pyrimidine metabolism, and protein processing in endoplasmic reticulum have been significantly enriched, while metabolomic analysis revealed a number of up- and downregulated DEMs found, which suggest that pyrimidine metabolism and fructose and mannose metabolism have been significantly enriched via metabolome. *pck2* and L-malic acid showed a highly significant positive correlation with the pathways of the pyruvate cycle and the citric acid cycle, which may further make impairment and ENR preservation in the liver.

## Figures and Tables

**Figure 1 animals-15-02962-f001:**
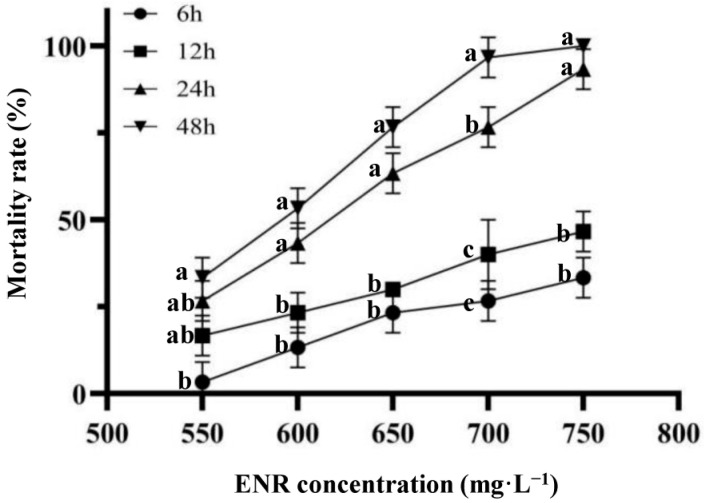
The mortality for ENR in American shad. Groups with different lowercase letters have significantly different mortality.

**Figure 2 animals-15-02962-f002:**
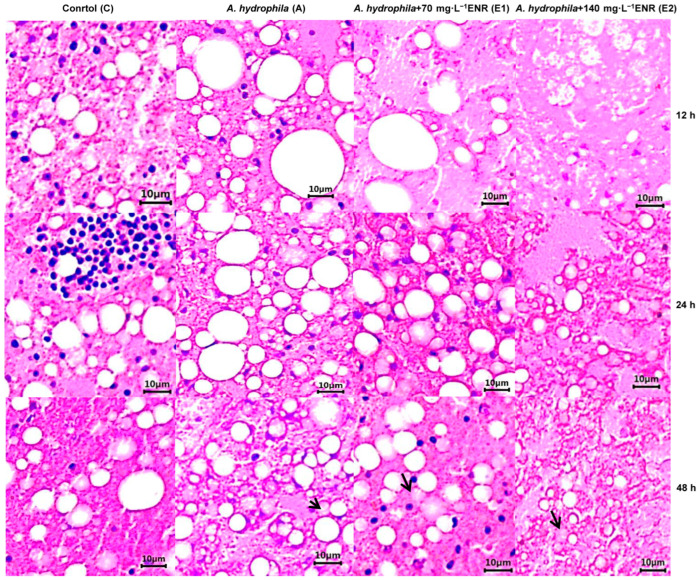
Liver histological changes in different treatment groups. The arrow indicates some liver cells atrophied at 48 h.

**Figure 3 animals-15-02962-f003:**
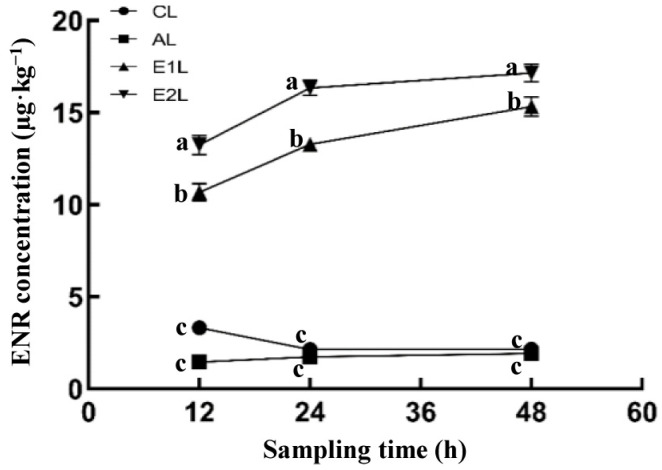
ENR concentrations in American shad muscle samples (mg·kg^−1^). The different lowercase letters stand for different significant levels. C, A, E1, and E2 stand for control, *A. hydrophila*, *A. hydrophila* + 70 mg·L^−1^ ENR, and *A. hydrophila* + 140 mg·L^−1^ ENR, respectively.

**Figure 4 animals-15-02962-f004:**
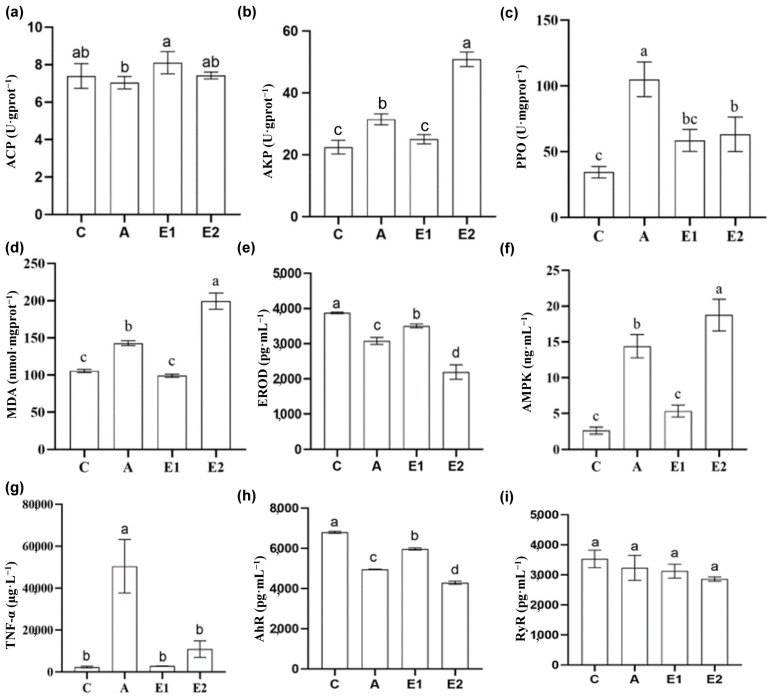
ENR exposure on hepatic enzymatic activities of *Alosa sapidissima*. C, A, E1, and E2 stand for control, *A. hydrophila*, *A. hydrophila* + 70 mg·L^−1^ ENR, and *A. hydrophila* + 140 mg·L^−1^ ENR, respectively. Alkaline phosphatase ACP (U·gprot^−1^), acid phosphatase AKP (U·gprot^−1^), polyphenol oxidase enzyme PPO (U·mgprot^−1^), trace malondialdehyde MDA (nmol·mgprot^−1^), acquired cytochrome p450 1a1 EROD (pg·mL^−1^), adenosine 5‘-monophosphate-activated protein kinase AMPK (µg·L^−1^), tumor necrosis factor α TNF-α (µg·L^−1^), aryl hydrocarbon receptor AhR (pg·mL^−1^), and ryanodine receptor RyR (pg·mL^−1^). The different lowercase letters stand for different significant levels.

**Figure 5 animals-15-02962-f005:**
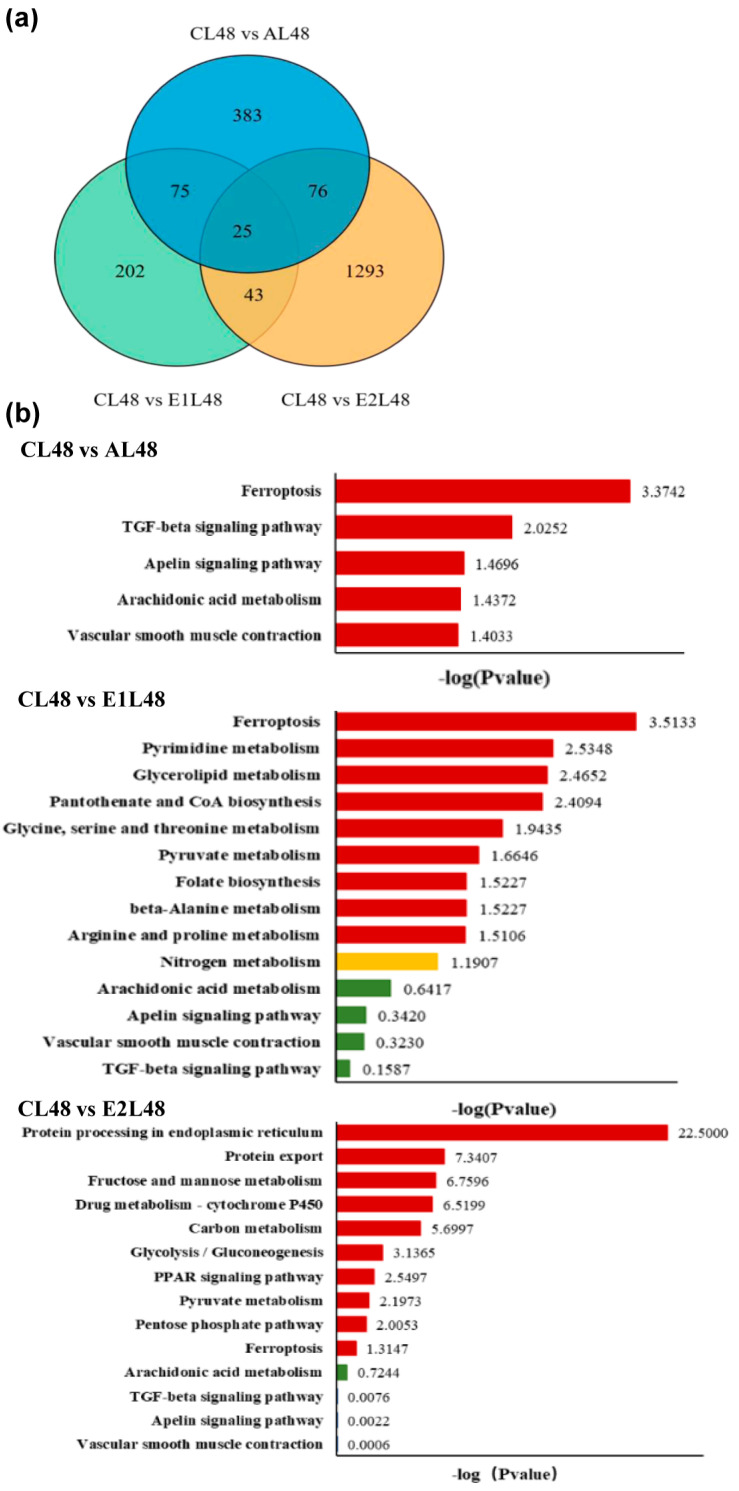
Venn diagram (**a**) of DEGs and enrichment pathways (**b**) between the treatment group and the control group. The comparison groups were CL48 vs. AL48, CL48 vs. E1L48, and CL48 vs. E2L48. “L” stands for liver samples, and 48 reveals 48 h. Red represents significant enrichment (*p* < 0.05), while the yellow and green colors represent non-significant with some enrichment.

**Figure 6 animals-15-02962-f006:**
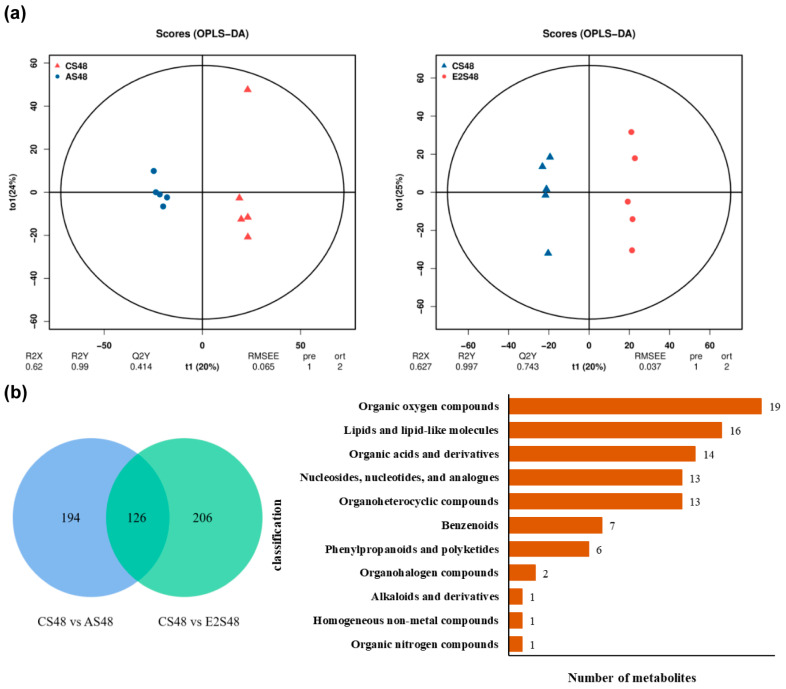
Sample repeatability (**a**), Venn diagram (**b**), enriched pathways (**c**), and OPLS-DA score map (**d**) of liver metabolome in CS48 vs. AS48 and CS48 vs. E2S48. (**a**), sample repeatability. (**b**), Venn diagram of DEMs and classification diagram of DEMs for CS48 vs. AS48 and CS48 vs. E2S48 showing with different KEGG pathway. (**c**,**d**), CS48 vs. AS48 and CS48 vs. E2S48 KEGG enrichment signaling pathways in OPLS-DA score map. “S” stands for liver metabolome samples, and 48 reveals 48 h. Red frame represents significant enrichment (*p* < 0.05).

**Figure 7 animals-15-02962-f007:**
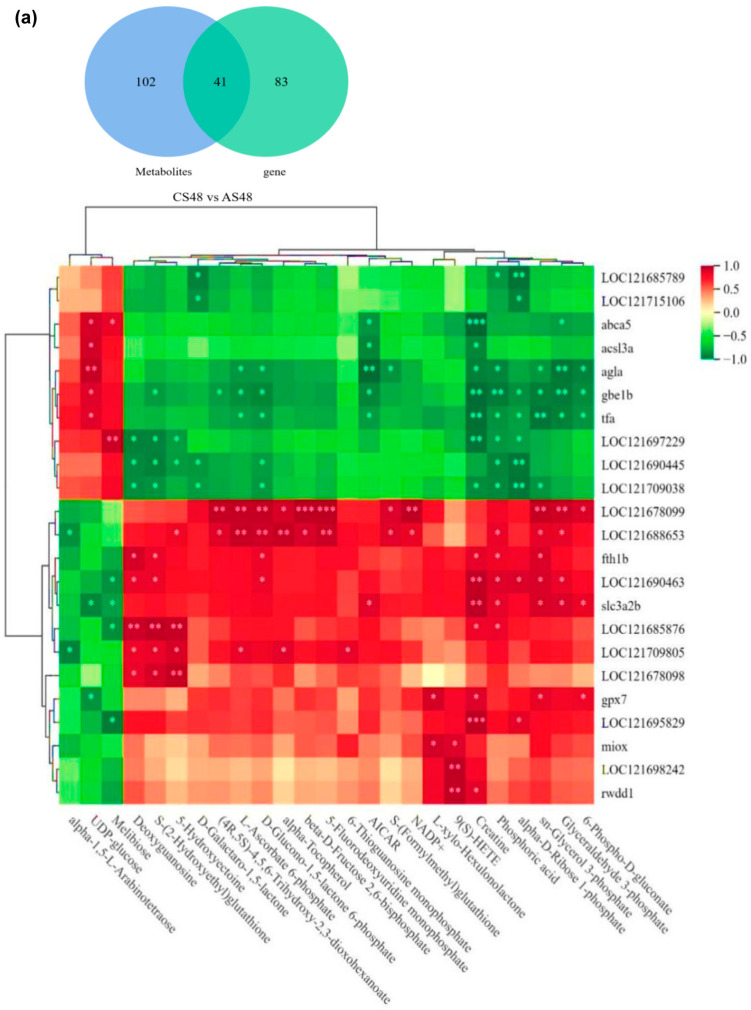
Correlation clustering heatmap for joint analysis of A vs. C and E2 vs. C groups. (**a**), CS48 vs. AS48. (**b**), CS48 vs. E2S48. (**a**), *loc121685789*, epoxide hydrolase 1-like, *loc121715106*, UDP-glucuronosyltransferase 2C1-like, *abca5*, ATP binding cassette subfamily A member 5, *acs13a*, Long-chain-fatty-acid--CoA ligase a, *agla*, amylo-alpha-1, 6-glucosidase, 4-alpha-glucanotransferase a, *gbelb*, un-annotated protein, *tfa*, transferrin-a, *loc121697229*, ribose-phosphate pyrophosphokinase 2, *loc121690445*, cellular tumor antigen p53-like, *loci21709038*, carbonyl reductase [NADPH] 1-like, *loci21678099*, cytochrome P450 2J6-like, *loc121688653*, lipocalin-like, *fthlb*, FlhB-like flagellar biosynthesis protein, *loc121690463*, L-serine dehydratase/L-threonine deaminase-like, *slc3a2b*, solute carrier family 3 member 2b, *loci21685876*, un-annotated protein, *loc121709805*, peroxisomal sarcosine oxidase-like, *loc121678098*, cytochrome P450 2J2-like, *gpx7*, glutathione peroxidase 7, *loci21695829*, un-annotated protein, *miox*, myo-inositol oxygenase, *loc121698242*, cytochrome b-245, beta polypeptide, *rwddl*, and RWD domain containing 4 like. (**b**), *loc121688100*, dihydrolipoyllysine-residue acetyltransferase component of pyruvate dehydrogenase complex, mitochondrial-like, *loc121694819*, ribose-phosphate pyrophosphokinase 1, *gale*, UDP-glucose 4-epimerase, *emppb*, GDP-mannose pyrophosphorylase B, *empnst1*, un-annotated protein, *adpgk2*, un-annotated protein, *pmm2*, phosphomannomutase 2, *acssl*, hsFATP2a_ACSVL_like domain-containing protein, *loc121705140*, citrate synthase, mitochondrial-like, *ptklh*, parathyroid hormone like hormone, *echsl*, chotochlorophyllide reductase subunit L, *gckr*, glucokinase regulator, *pith*, low-affinity phosphate transporter PitH, *amt*, aminomethyltransferase, *gludlb*, NAD-specific glutamate dehydrogenase, *loc121680065*, fructose-1,6-bisphosphatase 1-like, *got2b*, glutamic-oxaloacetic transaminase 2b, mitochondrial, *tkfe*, un-annotated protein, *aldob*, aldolase, fructose-bisphosphate B, *acss21*, un-annotated protein, *mdhlaa*, un-annotated protein, *ugt5dl*, un-annotated protein, *acoxl*, acyl-CoA oxidase like, *zgc_56622*, aldo/keto reductase, *idh2*, isocitrate dehydrogenase (NADP(+)) 2, *loc121681730*, UDP-glucuronosyltransferase 1-2-like, *pck2*, phosphoenolpyruvate carboxykinase 2, *loc121697229*, ribose-phosphate pyrophosphokinase 2, *cat*, catalase, *loc121690457*, aldehyde dehydrogenase, mitochondrial-like, *ugp2a*, and UDP-glucose pyrophosphorylase 2a. *, ** and *** stand for *p* < 0.05, *p* < 0.01, *p* < 0.001.

**Figure 8 animals-15-02962-f008:**
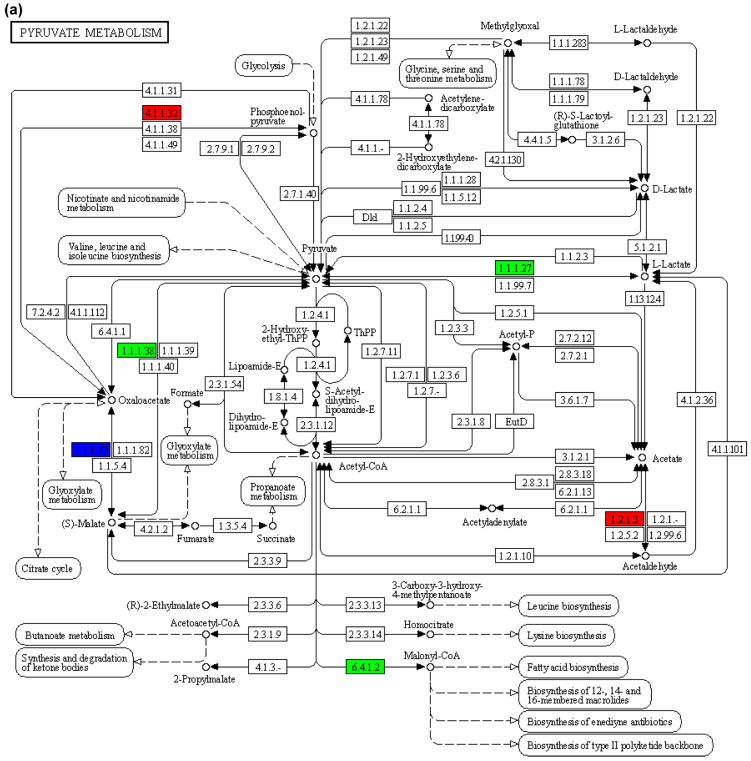
KEGG map of the citric acid cycle (TCA cycle (**a**), glycolysis/glucone ogeesis (**b**), and pyruvate metabolism (**c**)). Red and green stand for up- and down-regulation. (**a**), 4.1.1.32, phosphoenolpyruvate carboxykinase, 1.1.1.37, Malate dehydrogenase, 2.3.3.1, Citrate synthase, 2.3.3.8, putative ATP citrate synthase, and 1.1.1.41, isocitrate dehydrogenase (NAD). (**b**), 5.4.2.2, Phosphoglucomutase, 2.7.1.1, hexokinase, 5.3.1.9, Glucose-6-phosphate isomerase, 2.7.1.11, 6-phosphofructokinase, 4.1.2.13, Fructose-bisphosphate aldolase, 5.3.1.1, Triosephosphate isomerase, 1.2.1.12, Glyceraldehyde 3-phosphate dehydrogenase, 4.2.1.11, Enolase, 1.1.1.27, L-lactate dehydrogenase, 1.2.1.3, Aldehyde dehydrogenase, 1.1.1.1, Alcohol dehydrogenase, and 1.2.1.5, aldehyde dehydrogenase. (**c**), 1.1.1.38, NAD-dependent malic enzyme and 6.4.1.2, Acetyl-coenzyme A carboxylase carboxyl transferase subunit beta. Red and green represent up- and down-regulations, while blue stands for up- and down-regulations co-existing with bidirectional regulation under a certain condition.

**Table 1 animals-15-02962-t001:** Selected DEGs and qPCR verification.

Name	CL48 vs. AL48			CL48 vs. E1L48			CL48 vs. E2L48		
	Log2FC	*p* Value	qPCR	Log2FC	*p* Value	qPCR	Log2FC	*p* Value	qPCR
*loc121678099*	3.14	1.91 × 10^−7^	2.78	3.01	2.67 × 10^−5^	3.16	2.34	0.0002	2.55
*loc121684913*	−1.20	0.0041	−1.14	−1.19	0.0038	−1.28	1.25	0.0006	1.17
*loc121698291*	−1.54	8.78 × 10^−5^	−1.36	−3.86	7.08 × 10^−16^	−4.05	−1.88	0.0002	−2.02
*loc121699800*	−3.04	2.49 × 10^−5^	−2.96	−2.14	0.0061	−2.29	−2.79	4.92 × 10^−6^	−3.05
*loc121700164*	1.45	0.0029	1.33	2.74	0.0023	2.86	−1.34	0.0001	−1.36
*loc121707498*	−1.51	8.76 × 10^−7^	−1.53	−1.08	0.0088	−1.09	−3.80	2.99 × 10^−28^	−3.94
*loc121709038*	−2.18	4.66 × 10^−6^	−2.34	−1.34	0.0001	−1.47	−1.49	0.0033	−1.52
*loc121709129*	−0.74	0.0039	−0.86	−0.70	0.0017	−0.82	−0.59	0.0089	−0.63
*chka*	1.10	8.60 × 10^−5^	1.21	1.21	0.0068	1.30	−0.83	0.0040	−0.92
*dnajb6b*	−0.83	0.0008	−0.96	−0.90	0.0005	−0.92	−0.77	0.0081	−0.82
*dnajc3a*	−0.83	0.0055	−0.78	−0.83	0.0091	−0.84	−2.00	1.36 × 10^−13^	−2.31
*efr3ba*	−1.24	0.0008	−1.34	−0.98	0.0053	−1.03	−0.90	0.0065	−1.06
*igfbp1a*	1.99	0.0007	2.01	2.35	0.0049	2.34	−2.04	4.91 × 10^−5^	−2.37
*klf10*	1.55	0.0028	1.62	2.47	0.0035	2.58	−2.06	6.08 × 10^−7^	−2.18
*nfil3-5*	−1.46	0.0001	−1.54	−0.89	0.0016	−0.94	−0.84	0.0037	−0.92
*sept4b*	1.56	0.0003	1.63	1.19	0.0001	1.23	1.70	1.05 × 10^−8^	1.77
*sfxn1*	−0.84	0.0010	−0.88	−1.19	0.0073	−1.17	−0.66	0.0052	−0.67
*si_dkey-34d22.1*	1.36	0.0007	1.34	1.63	0.0060	1.72	−1.04	0.0012	−1.19
*slc10a4*	1.60	0.0006	1.62	2.22	0.0059	2.27	−1.60	0.0035	−1.74
*slc3a2b*	1.00	4.67 × 10^−6^	1.17	1.16	0.0005	1.25	−1.15	5.99 × 10^−5^	−1.23
*sptb*	−1.01	0.0001	−1.09	−1.09	0.0048	−1.18	0.66	0.0074	0.67
*tfa*	−0.79	0.0004	−0.84	−0.74	0.0092	−0.86	−0.73	4.60 × 10^−5^	−0.82
*tgfb2*	−1.38	1.06 × 10^−5^	−1.34	−1.14	3.08 × 10^−5^	−1.27	−1.45	9.88 × 10^−6^	−1.55
*vkorc1*	−0.74	0.0064	−0.82	−1.10	0.0021	−1.16	−1.31	1.62 × 10^−6^	−1.39

Note: “L” stands for liver samples, and 48 reveals 48 h. *loc121678099*, cytochrome P450 2J6-like, *loc121684913*, plectin-like, *loc121698291*, probable serine/threonine-protein kinase SIK1B, *loc121699800*, transmembrane protease serine 9-like, *loc121700164*, tumor necrosis factor receptor superfamily member 12A, *loc121707498*, complement C1q-like protein 2, *loc121709038*, carbonyl reductase [NADPH] 1-like, *loc121709129*, mediator of RNA polymerase II transcription subunit 13-like, *chka*, choline kinase alpha, *dnajb6b*, DnaJ heat shock protein family (Hsp40) member B6b, *dnajc3a*, DnaJ (Hsp40) homolog, subfamily C, member 3a, *efr3ba*, EFR3 homolog Ba, *igfbp1a*, insulin-like growth factor binding protein 1a, *klf10*, Kruppel-like transcription factor 10, *nfil3-5*, nuclear factor, interleukin 3 regulated, member 5, *sept4b*, septin 4b, *sfxn1*, sideroflexin 1, *si_dkey-34d22.1*, discoidin, CUB and LCCL domain-containing protein 1, *slc10a4*, solute carrier family 10 (sodium/bile acid cotransporter family), member 4, *slc3a2b*, solute carrier family 3 member 2b, *sptb*, spectrin beta, erythrocytic, *tfa*, transferrin-a, *tgfb2*, transforming growth factor beta 2, *vkorc1*, vitamin K epoxide reductase complex subunit 1.

**Table 2 animals-15-02962-t002:** The number of DEGs and DEMs in the KEGG pathway significantly enriched in the C and E2 groups.

KEGG Pathway	Gene Number	Metabolite Number	Ko ID
Fructose and mannose metabolism	16	5	ko00051
Carbon metabolism	31	7	ko01200
Ascorbate and aldarate metabolism	10	4	ko00053
Glyoxylate and dicarboxylate metabolism	11	4	ko00630
Pentose and glucuronate interconversions	13	3	ko00040
Citrate cycle (TCA cycle)	10	3	ko00020
Pyruvate metabolism	11	3	ko00620
Amino sugar and nucleotide sugar metabolism	17	6	ko00520

## Data Availability

Additional data supporting the findings of this study are available from the corresponding author upon reasonable request.

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
