# Peer review of "Analysis of the Biochemical Effect of Enrofloxacin on American Shad (Alosa sapidissima) Infected with Aeromonas hydrophila"

_animals, 2025, doi:10.3390/ani15202962_

Round 1

Reviewer 1 Report

Comments and Suggestions for Authors

I am not so fond of the title, as it says they have investigated therapeutic effects of ENR. But the therapeutic effect for an antibiotic is the bacteriocidal or bacteriostatic effect. Thus a more correct phrasing would instead be “biological effects” or “biochemical effects” or “metabolic effects”.

Simple summary

In the Simple summary, it is stated that the study is about the toxic effects of ENR “after 30 days” and this is repeated in the introduction. Yet, there is no evidence in the rest of the manuscript that the effect has been measured 30 days post exposure, instead it is stated that samples were taken after 12, 24 and 48 hours.

Please remove the sentence about Chinese herbs, that is completely irrelevant in this context.

Abstract

Line 34. “The addition of ENR in aquaculture significantly improved liver metabolic abnormalities caused by A. hydrophila”. What do you mean by improve? Did it enhance (increase) or mitigate (dampen) the effects? Chose another word that clearly states what happens.

Line 41-46: How do you rule out that the effects you look at are actually caused by A. hydrophila? Since the same effects are seen with treatment, the effects of A. hydrophila must be taken into account, even if there is some difference between the treatment groups. Thus, your statement about excessive use of ENR and the effects cannot be completely supported by your data. A treatment group just receiving ENR would have been relevant.

A lot of the manuscript is dedicated to the metabolomic effects, yet nothing is mentioned about this in the abstract. Please shorten other parts to get something about the metabolomics to fit in here.

Introduction

Lines 55-56. The reference [1] is from 1995 and about shad embryology, it is not relevant for the market value of shad in 2025. Please provide a propriate reference.

Lines 57-59. The sentence about collagen extraction has a reference [5] about oomycete pathogens. That seems quite irrelevant to the subject. The same reference is used on the next line about shad susceptibility to infections, but the reference is about Mugil cephalus. Please provide two other references for these statements.

I have not checked all references thoroughly because it took a lot of time to work through the manuscript, please make sure that all references are in order.

Lines 63-64. Please provide a reference that proves A. hydrophila is the most common freshwater pathogen, and also where it is most common (China, Asia, the whole world?).

Lines 97-106. Why would this be relevant only to shad? It would be better to write from a general “fish” or aquaculture perspective and refer to shad when there are specific studies about the species.

Materials and methods

Paragraph headers need to be rephrased to fit the content (order of the methods), eg paragraph header 2.1 – switch to “Experimental setup and sampling”

 please make sure the text is in proper sentences and proper tense instead of a bullet point/protocol like shape.

2.1

General comments

The paragraph lacks information on shad acclimatization time from purchase to the start of each trial.

Why was ENR exposure done through water instead of through feed, when per os is the  normal administration route for antibiotics in aquaculture?

How long before ENR treatment were the fish infected with A. hydrophila? This needs to be stated.

Was there continuous administration of ENR, or ENR supplement at any time during the challenge to the water to compensate for the ENR that was absorbed by the fish? Otherwise, they will not have been exposed to 70 and 140 mg/L for 12, 24 or 48 h.

Is the ENR exposure levels relevant under normal farming conditions? What is the dosage used to treat shad? According to what I can find (we do not use ENR to treat fish in my country) dosage for ornamental fish is 2.5-5 mg/L, which is way under what has been used in this study.

The reason for performing the LC50 test is not clearly stated. Was it used to calculate a maximum safe concentration for the other experiment? Right now it is just an extra text in the MM and results sections that don’t seem to fit a purpose in relation to the large experiment (no samples taken, no in-depth analyses) and if not justified for the other experiment I think it should be removed completely from the manuscript.

Line 121 “for A. hydrophila exposure and ENR treatment test” is unnecessary information, delete.

Line 132 “feeding and body weight changes were adjusted accordingly”. Do you mean “feeding was adjusted according to body weight changes?

Line 147. “The basic information of A. hydrophila used in this experiment was followed”. This is a very strange sentence. Do you mean that you followed a protocol from the reference provided? Please re-phrase.

Line 154 and line 165-166. You mention a feeding trial/test in these lines, but there is nothing else in the manuscript that indicates that there has been a feeding trial, so please delete “feeding”.

Line 154 “slightly paralyzed”. Do you mean they were sedated or under anaesthesia? Please also state here the drug used (probably MS-222 since that is mentioned later), dosage and approximate exposure time.

Line 155. Infusions are done intravenously or intra-arterially. The fish were injected

Line 155-156. Please remove the additional sentence about sedation/anaesthesia and i.p. injection as that was stated in the previous sentences and just confirm that the fish was returned to the holding tanks.

Lines 158-163. You mention twice that you have 20 fish in each tank and three/triplicate tanks per group.

Lines 163-170. The sampling regime (liver for HE etc) is basically described twice. I would suggest you remove the sentence on line 163-164 and keep the more detailed one on line 168-169.

Line 166-168. Here “L” is added to the group names (e.g. CL12). “L” needs to be explained here (not suddenly appear as a footnote to a figure text) if it is to be used. However, since you have only sampled liver, I think it just complicates the text/adds some confusion, and that you should stick to C12 etc.

Line 170. Please state the total number of samples per treatment group (triplicate tanks would mean 9 samples (HE, RNA-seq, qPCR) and 15 samples (metabolome) per treatment group and exposure time right)? Also see next comment, because I am not at all sure about this assumption.

Line 170 “Fish were starved for 24 hours before sampling”. That would mean the fish sampled after 12 h were starved for 12 h pre-exposure, the fish sampled at 24 h were starved from exposure and the fish sampled at 48 h were starved from 24 h post exposure? This would mean that there is an additional “treatment” factor that could affect the metabolism pathways differently in the fish. This also means that all 12 h fish per treatment group needs to be in one tank etc. This in turn would mean that you don’t have triplicates per treatment, because there is just one tank per treatment.

Could you please clarify about the starvation and whether repeated sampling (12, 24, 48 h) was performed per tank or if each exposure time had its separate tank.

Line 172. Samples cannot have been collected under anaesthesia, fish must have been euthanized. Please, also state that this was done using and overdose of MS-222, as I understand this was the case.

Line 174. Why was the fish frozen? Was it for ENR residues in muscle tissue? This is not clearly described.

2.2 – 2.4

Line 177. “of each group” should be “of each tank”?

Line 182. Please state the magnifications used to investigate the tissues.

Line 213. Remove “blood” or you need to state where you got the blood from in the sampling paragraph

Line 238. You state twice in the same sentence that 5 samples per group were used. Please re-phrase.

Results

Paragraph headers need to be rephrased to fit the content (order of the results), eg paragraph header 3.1 – ENR concentrations needs to be mentioned first if written as currently is, or this should be divided into two separate paragraphs. Histopathology would fit best as the first paragraph, as ENR concentrations are more related to the other microbiological analyses performed.

Be consistent in using past tense to describe the results.

Table 1 and figure 6 footnotes. Tables and figures should be self-explanatory, but in this case, with all abbreviations of genes, the footnotes become massive and disrupt the text. My suggestion, if the Editorial board allows that, is that you refer to a appendix/supplementary file where all abbreviations for the metabolomics can be gathered and explained.

For the LC50 test, I refer to my previous comment about the analysis seeming unrelated to the in-depth biological/physiological/metabolomic analyses performed and described in the manuscript.

Similarly to the LC50 test, the muscle tissue analysis seems out of scope here, as there are just small chunks of text (MM and results) that are not really related to the toxicity analyses performed on liver samples. The relevance of the muscle tissue tests for this specific study should be stated or else the text parts should be deleted. If to be kept, you also have to argue about where the ENR came from in the non-ENR-treated groups in the discussion. Are they in line with what can be accumulated in the fish from water residues?

Figure 1. “The different lowercase letters…” There are no lowercase letters in the figures.

Line 282-283 “various tissues and organs…. Showed a series of damages and lesions”. According to the MM section, only liver was sampled for histopathology (HP). Thus you need to rephrase here to “the liver…” or state in the MM section which other tissues were investigated by HP and also present clear results about that.

Line 285-286. “Liver cells showed….” Please state if you mean the control group. Please make a correct reference to figure 2, there is no “a” in that figure. What do you mean by stacking?

Lines 287-296. Please use proper medical language when referring to the morphological alterations seen in histopathology. What do you mean by “enlarged”? Simple hypertrophy or did they swell due to massive vacuoles forming? “Watery” is not correct. Do you mean liquefactive? What do you mean by “cell accumulation?

Figure 2: Histopathology photos are blurry - nuclei are not sharp, they all look a bit condensed/atrophic, cell borders cannot be properly seen. “Control 24 h” actually looks as bad as “E2 24 h” (no obvious nuclei, blurry purple staining between the vacuoles).  There are preparation artefacts (cracks) in the control group photos. All slides are either overstained with eosin or the pink coloration has been intensified. A paler pink coloration would enhance interpretation. Furthermore, the scale bar sizes reveal that the figures are not at equal scale, so comparing the size of vacuoles etc is a bit hard. One of the arrows (A 48 h) seem to point at a nucleus that is compressed due to extreme vacuolization, this is also potentially present in one of the control group slides (C 48 h). One arrow (E2 48 h) just seems to point at a vacuole. Magnification should be stated. The black “circule” in the E2 48 h is unnecessary as the whole slide has the same morphology, and it is just better described in the text. Patches of similar morphology seems to be present in some other slides from different groups as well.

3.2

The first sentence should be removed.

Please refer to the figure in the text, e.g. a “… at 48 h. The results can be seen in figure 3.”

Why is EROD only mentioned in relation to E1? It was significantly reduced in groups A and E2 as well.

Line 337-340. It would be nice if the number of genes without expression differences (according to Fig 4a n=only 25) were also stated.

Line 382. “S” is suddenly introduced in the definition of the treatment groups (CS48 etc). What does “S” stand for? This needs to be explained here, not in the Figure 5 text several pages later. Again (as with the “L” addition to the group definitions), I do not think it is necessary to use an extra letter to describe the samples, just stick with the original group definers C, A, E1 and E2 followed by sampling time. It is apparent that it is related to the metabolome.

Figure 4, footnote “Red represents significant enrichment (p<0.05)”. It would be nice with a statement of the yellow and green colors as well (apparently non-significant but still some enrichment?)

Figure 5d – the symbol explanation to the right has been compressed (compare to 5e).

Figure 6. For me that is not working with metabolomics, the figure is confusing. The color scale of the figure needs to be explained. In addition, red and green is not a good combination for color blinds, but I suppose colors are automatically generated by the software used. The figures are blurry. Is one treatment group on the “X axis” and one on the “Y axis” of the figure and in that case which one is where? Or how can you see what loci etc correlates to what treatment group? In 6a (at 150% size), it is barely possible to see that the “dots” in the squares are asterisks, and thus I suppose they represent values (1-3 asterisks). The “*” should be explained in the footnote, as well as the red phrame of L-malic acid and pck2. I refer to previous comment about collecting the explanation of the genes in a separate file, but in the footnot “unchactered” protein is mentioned several times. Do you mean “uncharted”?

Line 471. “Significantly enriched” how? In both groups, in one of the groups? This might be related to me not being into metabolomics as already mentioned, thus failing to interpret fig 6, but I think it could be clarified in the text. Someone else that is more interested in basic toxic or metabolic effects of antibiotics might be interested in this paper.

The whole paragraph that starts on line 471 is heavy to read as it just stacks a lot of genes, metabolites etc and p-values. Is it simply some of the results from figure 6 stated in text? In that case, would it be possible to just delete everything in the paragraph from line 474 (from “There was a..”) until line 508 “…ugp2a (P<0.01).” This would enhance readability. You could perhaps then highlight the chosen pathways in the figure and refer to that in the first sentence of the paragraph. If the text is kept, please check that relationships are only mentioned once (L-malic acid vs. pck2 with p<0.0001) is mentioned at least twice.

Line 509-510 Switch order of mentioning the TCA cycle and pyruvate cycle to reflect order in the figure (or switch order of figures). Also, for the same reason, it would be good if the glycolysis/gluconeogenesis was instead figure 7c.

Discussion

Start with your findings (line 536-539) instead of starting to discuss environmental antibiotics and then add your results. I would like to see a discussion about whether the effects of the 140 mg ENR treatment could actually be effects of the A. hydrophila infection rather than pure antibiotic effects prior to your discussion on the concerns about environmental ENR. How can you isolate the A. hydrophila and 140 mg ENR effects from each other when it comes to the enzyme activities only? Could a 140 mg ENR treatment group without A. hydrophila exposure have given that answer?

Please also in the discussion refer to whether the treatment dosage you used would be considered normal or toxic exposure, as this has bearing on your discussion on both the effects you have found and the fact that you see these effect as a problem for the fish and use this to further problematize the use of antibiotics. Don’t understand me wrong, antibiotic overuse, environmental residues AMR is a problem, but to argue for that based on your results, the results need to be valid under normal circumstances.

Lines 552-553. What do you mean by “has revealed focusing on elevated temperatures”?

Lines 557. “In this study…” ? Do you mean “in one/some study/studies”? You are clearly not referring to your own study as you have inserted references, and there are things mentioned that you have not looked at.

Lines 578-581. The sentence about the TCA cycle is really strange. You refer to the bacterium Gallibacterium anatis as being the main metabolic pathway…. Please rephrase the sentence.

Lines 600-602. You have performed a metabolomic study on fish livers, you have not looked at microbes or water. Please explain how your results could support the idea that microbial and algal preparations improve water quality and microbial diversity. In fact, the whole text from line 587-605 is quite out of the scope for your study. You should remove the header plus that text and just add the text from line 606 directly to the text that ends on line 586.

Conclusions

The conclusions are to detailed. Instead of repeating results, summarize them, like “Several liver enzymes….” And “Metabolomic analysis revealed a number of up- and downregulated…”

Lines 617-618. The statement about muscle tissue should be removed, the results are not mentioned in the discussion and can thus not be a significant conclusion of this study.

The last sentence is a bit out of the scope, see previous comments. 

Comments on the Quality of English Language

The language needs to be thoroughly checked by someone who speaks and writes English fluently

There is a distinct difference in the introduction (fluent, AI generated?) compared to the simple summary, abstract and materials and methods, whereas the results and discussion are OK but still can be improved. 

The MM section basically is like a bullet point protocol where the bullet points have been lined up to create pseudo-sentences. 

There are several strange words, like "bucket" when you mean a fish tank. 

Tense should be checked throughout the manuscript. 

Author Response

English Language and Figures

Reviewer 1 Round 1 The English could be improved to more clearly express the research.

Comments on the Quality of English Language

The language needs to be thoroughly checked by someone who speaks and writes English fluently. There is a distinct difference in the introduction (fluent, AI generated?) compared to the simple summary, abstract and materials and methods, whereas the results and discussion are OK but still can be improved. The MM section basically is like a bullet point protocol where the bullet points have been lined up to create pseudo-sentences. There are several strange words, like "bucket" when you mean a fish tank. Tense should be checked throughout the manuscript. 

Response: Thank you for the suggestions. The revised document has been polished by English native speaker Ampeire Yona, Hundo Rumuri Victor and Frank Nzeyimana. The MM section has been revised and “bucket” changed in line 123, also the tense has been revised.

Quality of Figures

Reviewer 1 Round 1 Figures and tables can be improved

Response: Thank you for the suggestions. Figures (1-3, 4b, 6d, 8) have been changed, while tables not changed because we have tried our best.

Reviewer#1

I am not so fond of the title, as it says they have investigated therapeutic effects of ENR. But the therapeutic effect for an antibiotic is the bacteriocidal or bacteriostatic effect. Thus a more correct phrasing would instead be “biological effects” or “biochemical effects” or “metabolic effects”.

Response: Thank you, it has been changedchanged to “biochemical” in line 2, 13.

Simple summary

In the Simple summary, it is stated that the study is about the toxic effects of ENR “after 30 days” and this is repeated in the introduction. Yet, there is no evidence in the rest of the manuscript that the effect has been measured 30 days post exposure, instead it is stated that samples were taken after 12, 24 and 48 hours.

Response: Thank you, “after 30 days” changed to “for 12~48 h” in Simple summary and line 115.

Please remove the sentence about Chinese herbs, that is completely irrelevant in this context.

Response: Thank you for the suggestions. The sentence about Chinese herbs has been deleted. Then the simple summary as “American shad’s culture is increasingly threatened by the potential bacteria and cured by the widespread use of antibiotics, like enrofloxacin (ENR). The present study addresses these issues to analyze the toxic effects of ENR exposure on American shad for 12~48 h, particularly on metabolic changes using transcriptomics and metabolomics. The fluctuated enzymatic activities revealed, the pathways including ferroptosis, pyrimidine metabolism, protein processing in endoplasmic reticulum have been significantly enriched, and the key metabolite, L-malic acid showed a highly significant positive correlation with the pathway of pyruvate cycle and citric acid cycle.”.

Abstract

Line 34. “The addition of ENR in aquaculture significantly improved liver metabolic abnormalities caused by A. hydrophila”. What do you mean by improve? Did it enhance (increase) or mitigate (dampen) the effects? Chose another word that clearly states what happens.

Response: Thank you for the suggestions, “enhanced” added in line 31.

Line 41-46: How do you rule out that the effects you look at are actually caused by A. hydrophila? Since the same effects are seen with treatment, the effects of A. hydrophila must be taken into account, even if there is some difference between the treatment groups. Thus, your statement about excessive use of ENR and the effects cannot be completely supported by your data. A treatment group just receiving ENR would have been relevant.

Response: Thank you for the suggestions, “drugs” changed to “chemical” in line 14, and “+A. hydrophila” has been added in line 16.

A lot of the manuscript is dedicated to the metabolomic effects, yet nothing is mentioned about this in the abstract. Please shorten other parts to get something about the metabolomics to fit in here.

Response: Thank you for the suggestions, the results about metabomics have been added in line 24-29, as “A total of 126 shared metabolites were found in the comparisons of A vs C, E2 vs C, and the main enriched pathway were organic oxygen compounds, lipids and lipid-like molecules. Except for fluorobenzoate degradation, the pathways of ascorbate and aldarate metabolism, pyrimidine metabolism were significantly increased in A and E2 groups, which further resulted in vacuolization, cell shedding and necrosis in the liver.”.

Introduction

Lines 55-56. The reference [1] is from 1995 and about shad embryology, it is not relevant for the market value of shad in 2025. Please provide a propriate reference.

Response: Thank you for the suggestions, the place for the cited reference [1] was changed in line 40.

Lines 57-59. The sentence about collagen extraction has a reference [5] about oomycete pathogens. That seems quite irrelevant to the subject. The same reference is used on the next line about shad susceptibility to infections, but the reference is about Mugil cephalus. Please provide two other references for these statements.

Response: Thank you for the suggestions, the reference of collagen extraction was deleted in line 46, 670, and reference for disease infections was changed in line 45.

I have not checked all references thoroughly because it took a lot of time to work through the manuscript, please make sure that all references are in order.

Response: Thank you for the suggestions, the references have been checked and updated. We changed the references [5-6] and added [20, 22-23], and especially deleted our studies [39-41] in line 777-784.

Lines 63-64. Please provide a reference that proves A. hydrophila is the most common freshwater pathogen, and also where it is most common (China, Asia, the whole world?).

Response: Thank you for the suggestions, the reference [6] has been added in line 48, 672-673.

Lines 97-106. Why would this be relevant only to shad? It would be better to write from a general “fish” or aquaculture perspective and refer to shad when there are specific studies about the species.

Response: Thank you for the suggestions, “fish” has been added instead for “shad” in line 92-93.

Materials and methods

Paragraph headers need to be rephrased to fit the content (order of the methods), eg paragraph header 2.1 – switch to “Experimental setup and sampling”

Response: Thank you, corrected as suggested in line 118.

please make sure the text is in proper sentences and proper tense instead of a bullet point/protocol like shape.

Response: Thank you for the suggestions, the tense for the whole document has been checked and revised, i.e., in line 72.

2.1General comments

The paragraph lacks information on shad acclimatization time from purchase to the start of each trial.

Response: Thank you for the suggestions, the details for fish maintenance, feeding revealed in line 122-123, 125-128.

Why was ENR exposure done through water instead of through feed, when per os is the normal administration route for antibiotics in aquaculture?

Response: Thank you for the suggestions. In the field of aquaculture, ENR through water may be convenient for fish farmers, and in further studies, the research of ENR addition in the feed will be performed following the reviewer’s suggestions.

How long before ENR treatment were the fish infected with A. hydrophila? This needs to be stated.

Response: Thank you for the suggestions, A. hydrophila was injected (line 154) intraperitoneally (line 156) for shad, and “once per day’ has been added in line 157.

Was there continuous administration of ENR, or ENR supplement at any time during the challenge to the water to compensate for the ENR that was absorbed by the fish? Otherwise, they will not have been exposed to 70 and 140 mg/L for 12, 24 or 48 h.

Response: Thank you for the suggestions. ENR’s elimination half-life in fish can exceed 20 hours, the exposure duration was 48 h, “static” has been added in line 159, and the concentrations of ENR were not maintained (replenish to keep  a certain concentration) in this study following the field ENR usage in China. “acute” added in line 114, the longer-exposure duration will be performed in the future, which may use fresh ENR solutions to maintain the concentration to ensure the effect on shad.

Is the ENR exposure levels relevant under normal farming conditions? What is the dosage used to treat shad? According to what I can find (we do not use ENR to treat fish in my country) dosage for ornamental fish is 2.5-5 mg/L, which is way under what has been used in this study.

Response: Thank you for the suggestions. ENR may be forbidden in your country, and in China, it can be used with the minimum residue standard of 100 µg/kg in the market. The concentration of ENR for shad may be higher than other fish species, and some foreign countries, “The ENR concentrations were chosen based on earlier studies that applied 50~100 mg·L-1 ENR for microalgae [22-23].” has been added in line 165-166. “normal” and “excessive dosage” added for 70 and 140 mg·L-1 in the revised document to make the potential readers understand more clearly.

The reason for performing the LC50 test is not clearly stated. Was it used to calculate a maximum safe concentration for the other experiment? Right now it is just an extra text in the MM and results sections that don’t seem to fit a purpose in relation to the large experiment (no samples taken, no in-depth analyses) and if not justified for the other experiment I think it should be removed completely from the manuscript.

Response: Thank you for the suggestions. The aim for LC50 test was to find the safe concentration of ENR range for shad, and especially in the toxic exposure test, it was a conventional method for researchers. “For finding the range of ENR addition dosage” has been added in line 134.

Line 121 “for A. hydrophila exposure and ENR treatment test” is unnecessary information, delete.

Response: Thank you for the suggestions, corrected as suggested in line 120.

Line 132 “feeding and body weight changes were adjusted accordingly”. Do you mean “feeding was adjusted according to body weight changes?

Response: Thank you, this sentence has been deleted in line 131-132.

Line 147. “The basic information of A. hydrophila used in this experiment was followed”. This is a very strange sentence. Do you mean that you followed a protocol from the reference provided? Please re-phrase.

Response: Thank you, “following a previously published report [21]” has been added in line 148-149.

Line 154 and line 165-166. You mention a feeding trial/test in these lines, but there is nothing else in the manuscript that indicates that there has been a feeding trial, so please delete “feeding”.

Response: Thank you for the suggestions. “of the feeding trial” and “feeding” have been deleted.

Line 154 “slightly paralyzed”. Do you mean they were sedated or under anaesthesia? Please also state here the drug used (probably MS-222 since that is mentioned later), dosage and approximate exposure time.

Response: Thank you for the suggestions, “MS-222 (10 mg·L-1 for 15 min)” has been added in line 155.

Line 155. Infusions are done intravenously or intra-arterially. The fish were injected

Response: Thank you for the suggestions, “intraperitoneally” has been used in line 156.

Line 155-156. Please remove the additional sentence about sedation/anaesthesia and i.p. injection as that was stated in the previous sentences and just confirm that the fish was returned to the holding tanks.

Response: Thank you for the suggestions, “At the start of the experiment, the fish were intraperitoneally injected with” has been deleted in line 155-156.

Lines 158-163. You mention twice that you have 20 fish in each tank and three/triplicate tanks per group.

Response: Thank you for the suggestions. Fish number is important for reviewers to know the experimental design, and “20” was deleted.

Lines 163-170. The sampling regime (liver for HE etc) is basically described twice. I would suggest you remove the sentence on line 163-164 and keep the more detailed one on line 168-169.

Response: Thank you for the suggestions. It has been deleted in line 167-170 and time has been added in line 175.

Line 166-168. Here “L” is added to the group names (e.g. CL12). “L” needs to be explained here (not suddenly appear as a footnote to a figure text) if it is to be used. However, since you have only sampled liver, I think it just complicates the text/adds some confusion, and that you should stick to C12 etc.

Response: Thank you for the suggestions. Because the figures contain “L”, “Here, “L” indicates liver samples” has been added in line 172-173.

Line 170. Please state the total number of samples per treatment group (triplicate tanks would mean 9 samples (HE, RNA-seq, qPCR) and 15 samples (metabolome) per treatment group and exposure time right)? Also see next comment, because I am not at all sure about this assumption.

Response: Thank you for the suggestions. The sample numbers will be 3 not 9 (3*3 in triplicate) according to the academic rules without calculating the repeats.

Line 170 “Fish were starved for 24 hours before sampling”. That would mean the fish sampled after 12 h were starved for 12 h pre-exposure, the fish sampled at 24 h were starved from exposure and the fish sampled at 48 h were starved from 24 h post exposure? This would mean that there is an additional “treatment” factor that could affect the metabolism pathways differently in the fish. This also means that all 12 h fish per treatment group needs to be in one tank etc. This in turn would mean that you don’t have triplicates per treatment, because there is just one tank per treatment.

Could you please clarify about the starvation and whether repeated sampling (12, 24, 48 h) was performed per tank or if each exposure time had its separate tank.

Response: Thank you for the suggestions. It is our fault for 24 h starving before sampling, and “were starved for 24 h” and “then” have been deleted.

Line 172. Samples cannot have been collected under anaesthesia, fish must have been euthanized. Please, also state that this was done using and overdose of MS-222, as I understand this was the case.

Response: Thank you for the suggestions. “MS-222” was deleted.

Line 174. Why was the fish frozen? Was it for ENR residues in muscle tissue? This is not clearly described.

Response: Thank you for the suggestions. The sentence was deleted in line 179-181.

2.2 – 2.4 Line 177. “of each group” should be “of each tank”?

Response: Thank you, corrected as suggested in line 173.

Line 182. Please state the magnifications used to investigate the tissues.

Response: Thank you, “400X,” was added in line 189.

Line 213. Remove “blood” or you need to state where you got the blood from in the sampling paragraph

Response: Thank you, “blood,” was deleted in line 220.

Line 238. You state twice in the same sentence that 5 samples per group were used. Please re-phrase.

Response: Thank you, (n=5) was deleted in line 246.

Results Paragraph headers need to be rephrased to fit the content (order of the results), eg paragraph header 3.1 – ENR concentrations needs to be mentioned first if written as currently is, or this should be divided into two separate paragraphs. Histopathology would fit best as the first paragraph, as ENR concentrations are more related to the other microbiological analyses performed.

Response: Thank you for the suggestions. The authors set histopathological results as the second paragraph in line 286, the paragraph about ENR has been divided into two separate paragraphs in line 275 and line 300.

Be consistent in using past tense to describe the results.

Response: Thank you for the suggestions. The authors revised the past tense, i.e., line 277.

Table 1 and figure 6 footnotes. Tables and figures should be self-explanatory, but in this case, with all abbreviations of genes, the footnotes become massive and disrupt the text. My suggestion, if the Editorial board allows that, is that you refer to a appendix/supplementary file where all abbreviations for the metabolomics can be gathered and explained.

Response: Thank you for the suggestions. The authors contacted with the editor, the editor said “Dear Dr. Zheng, Thank you for your email. You can include this reply in your point-to-point response to the reviewer. We will also attach this comment to the academic editor and ask him to check whether it is feasible.”. The footnotes for Table 1, Fig. 7 may be convenient for the readers to follow the key genes that they want to know. The authors ask the reviewer for help to receive this format.

For the LC50 test, I refer to my previous comment about the analysis seeming unrelated to the in-depth biological/physiological/metabolomic analyses performed and described in the manuscript.

Response: Thank you for the suggestions. The introduction (line 115), result (line 275-282) and discussion (line 537-538), which were relative with LC50 test has been revised, and “for finding the range of ENR addition dosage” has been added in line 134.

Similarly to the LC50 test, the muscle tissue analysis seems out of scope here, as there are just small chunks of text (MM and results) that are not really related to the toxicity analyses performed on liver samples. The relevance of the muscle tissue tests for this specific study should be stated or else the text parts should be deleted. If to be kept, you also have to argue about where the ENR came from in the non-ENR-treated groups in the discussion. Are they in line with what can be accumulated in the fish from water residues?

Response: Thank you for the suggestions. The ENR residue was found in the muscle of shad, even in the control groups, because ENR has half-life, its relative introduction (line 79), MM (line 190) and discussion (line 552-554) has been elaborated.

Figure 1. “The different lowercase letters…” There are no lowercase letters in the figures.

Response: Thank you for the suggestions. The authors have re-written the significant levels using different lowercase letters in Fig. 1 in line 284-285.

Line 282-283 “various tissues and organs…. Showed a series of damages and lesions”. According to the MM section, only liver was sampled for histopathology (HP). Thus you need to rephrase here to “the liver…” or state in the MM section which other tissues were investigated by HP and also present clear results about that.

Response: Thank you, corrected as suggested in line 288-295.

Line 285-286. “Liver cells showed….” Please state if you mean the control group. Please make a correct reference to figure 2, there is no “a” in that figure. What do you mean by stacking?

Response: Thank you, “In the control groups,” has been added in line 288, “a” and “stacking,” has been deleted.

Lines 287-296. Please use proper medical language when referring to the morphological alterations seen in histopathology. What do you mean by “enlarged”? Simple hypertrophy or did they swell due to massive vacuoles forming? “Watery” is not correct. Do you mean liquefactive? What do you mean by “cell accumulation?

Response: Thank you for the suggestions. It was “enlarged” by ENR exposure when compared with the controls. “Watery” and “cell accumulation” have been deleted in line 295.

Figure 2: Histopathology photos are blurry - nuclei are not sharp, they all look a bit condensed/atrophic, cell borders cannot be properly seen. “Control 24 h” actually looks as bad as “E2 24 h” (no obvious nuclei, blurry purple staining between the vacuoles).  There are preparation artefacts (cracks) in the control group photos. All slides are either overstained with eosin or the pink coloration has been intensified. A paler pink coloration would enhance interpretation. Furthermore, the scale bar sizes reveal that the figures are not at equal scale, so comparing the size of vacuoles etc is a bit hard. One of the arrows (A 48 h) seem to point at a nucleus that is compressed due to extreme vacuolization, this is also potentially present in one of the control group slides (C 48 h). One arrow (E2 48 h) just seems to point at a vacuole. Magnification should be stated. The black “circule” in the E2 48 h is unnecessary as the whole slide has the same morphology, and it is just better described in the text. Patches of similar morphology seems to be present in some other slides from different groups as well.

Response: Thank you for the suggestions. The result presentation in line 288-295, figure has been revised in line 296, and black circle has been removed.

3.2 The first sentence should be removed. Please refer to the figure in the text, e.g. a “… at 48 h. The results can be seen in figure 3.” Why is EROD only mentioned in relation to E1? It was significantly reduced in groups A and E2 as well.

Response: Thank you for the suggestions. The first sentence in 3.4 has been deleted in line 310, “(Fig. 4)” added in line 311, results about EROD (” A, , E2 s”) have been added in line 320.

Line 337-340. It would be nice if the number of genes without expression differences (according to Fig 4a n=only 25) were also stated.

Response: Thank you for the suggestions, “23478 annotated genes showed no significant difference in this study” has been added in line 340-341.

Line 382. “S” is suddenly introduced in the definition of the treatment groups (CS48 etc). What does “S” stand for? This needs to be explained here, not in the Figure 5 text several pages later. Again (as with the “L” addition to the group definitions), I do not think it is necessary to use an extra letter to describe the samples, just stick with the original group definers C, A, E1 and E2 followed by sampling time. It is apparent that it is related to the metabolome.

Response: Thank you for the suggestions, ““S” stands for metabolome samples” has been added in line 386, and “L” and “S” used to identify the samples for liver used in RNA-seq and metabolome.

Figure 4, footnote “Red represents significant enrichment (p<0.05)”. It would be nice with a statement of the yellow and green colors as well (apparently non-significant but still some enrichment?)

Response: Thank you for the suggestions, it has been added in line 408-409.

Figure 5d – the symbol explanation to the right has been compressed (compare to 5e).

Response: Thank you for the suggestions, the Fig.6d has been revised in line 412.

Figure 6. For me that is not working with metabolomics, the figure is confusing. The color scale of the figure needs to be explained. In addition, red and green is not a good combination for color blinds, but I suppose colors are automatically generated by the software used. The figures are blurry. Is one treatment group on the “X axis” and one on the “Y axis” of the figure and in that case which one is where? Or how can you see what loci etc correlates to what treatment group? In 6a (at 150% size), it is barely possible to see that the “dots” in the squares are asterisks, and thus I suppose they represent values (1-3 asterisks). The “*” should be explained in the footnote, as well as the red phrame of L-malic acid and pck2. I refer to previous comment about collecting the explanation of the genes in a separate file, but in the footnot “unchactered” protein is mentioned several times. Do you mean “uncharted”?

Response: Thank you for the suggestions. The figures were automatically generated by the software, and “unchartered” means “uncharted or un-annotated” because of not rich genomic data for the public.

Line 471. “Significantly enriched” how? In both groups, in one of the groups? This might be related to me not being into metabolomics as already mentioned, thus failing to interpret fig 6, but I think it could be clarified in the text. Someone else that is more interested in basic toxic or metabolic effects of antibiotics might be interested in this paper.

Response: Thank you for the suggestions, it means enriched in different comparison groups, and “further analyzed based on the selected…from the comparison” has been added in line 471-472.

The whole paragraph that starts on line 471 is heavy to read as it just stacks a lot of genes, metabolites etc and p-values. Is it simply some of the results from figure 6 stated in text? In that case, would it be possible to just delete everything in the paragraph from line 474 (from “There was a..”) until line 508 “…ugp2a (P<0.01).” This would enhance readability. You could perhaps then highlight the chosen pathways in the figure and refer to that in the first sentence of the paragraph. If the text is kept, please check that relationships are only mentioned once (L-malic acid vs. pck2 with p<0.0001) is mentioned at least twice.

Response: Thank you for the suggestions, the repeated parts have been checked and deleted, i.e., line 496-501 about pck2.

Line 509-510 Switch order of mentioning the TCA cycle and pyruvate cycle to reflect order in the figure (or switch order of figures). Also, for the same reason, it would be good if the glycolysis/gluconeogenesis was instead figure 7c.

Response: Thank you for the suggestions, it has been changed in line 516-518.

Discussion Start with your findings (line 536-539) instead of starting to discuss environmental antibiotics and then add your results. I would like to see a discussion about whether the effects of the 140 mg ENR treatment could actually be effects of the A. hydrophila infection rather than pure antibiotic effects prior to your discussion on the concerns about environmental ENR. How can you isolate the A. hydrophila and 140 mg ENR effects from each other when it comes to the enzyme activities only? Could a 140 mg ENR treatment group without A. hydrophila exposure have given that answer?

Response: Thank you for the suggestions. The discussion about therapeutic effect by ENR has been added in line 537-542. Actually, it should have the control group of ENR only (without A. hydrophila), the discussion about ENR effect has been added, however, the difference for the effect of ENR (both 70 and 140 mg·L-1) also is not clear even when we set a ENR only control group, because we used two concentrations of ENR.

Please also in the discussion refer to whether the treatment dosage you used would be considered normal or toxic exposure, as this has bearing on your discussion on both the effects you have found and the fact that you see these effect as a problem for the fish and use this to further problematize the use of antibiotics. Don’t understand me wrong, antibiotic overuse, environmental residues AMR is a problem, but to argue for that based on your results, the results need to be valid under normal circumstances.

Response: Thank you for the suggestions. The discussion about normal/toxic ENR exposure has been added in line 538-540, as “With normal ENR (group E1), the detected enzymatic activities except for EROD, AhR, showed no significant difference when compared with controls”.

Lines 552-553. What do you mean by “has revealed focusing on elevated temperatures”?

Response: Thank you, “the effect of… on American shad” has been added in line 559-560.

Lines 557. “In this study…” ? Do you mean “in one/some study/studies”? You are clearly not referring to your own study as you have inserted references, and there are things mentioned that you have not looked at.

Response: Thank you for the suggestions, “(which was demonstrated in shad from the reference…)” has been added in line 565.

Lines 578-581. The sentence about the TCA cycle is really strange. You refer to the bacterium Gallibacterium anatis as being the main metabolic pathway…. Please rephrase the sentence.

Response: Thank you for the suggestions, it has been changed to citric acid cycle pathway in line 587.

Lines 600-602. You have performed a metabolomic study on fish livers, you have not looked at microbes or water. Please explain how your results could support the idea that microbial and algal preparations improve water quality and microbial diversity. In fact, the whole text from line 587-605 is quite out of the scope for your study. You should remove the header plus that text and just add the text from line 606 directly to the text that ends on line 586.

Response: Thank you for the suggestions. This paragraph has been deleted in line 595, 608-613, 615-618.

Conclusions The conclusions are to detailed. Instead of repeating results, summarize them, like “Several liver enzymes….” And “Metabolomic analysis revealed a number of up- and downregulated…”

Response: Thank you for the suggestions, the conclusion has been revised in line 629, 633-634.

Lines 617-618. The statement about muscle tissue should be removed, the results are not mentioned in the discussion and can thus not be a significant conclusion of this study.

Response: Thank you for the suggestions, the conclusion about muscle was deleted in line 625-626.

The last sentence is a bit out of the scope, see previous comments. 

Response: Thank you for the suggestions, this sentence was deleted in line 638-640.

Reviewer 2 Report

Comments and Suggestions for Authors

The present study addresses these issues to analyze the toxic effects of ENR exposure on American shad after 30 days, particularly on metabolic changes using transcriptomics and metabolomics. The article provides detailed content and clear logic. However, here are some problems that need to be appropriately addressed in the article. Particular attention should be paid to substantial improvement in writing proficiency.

  1. In Abstract, the Latin name of American shad should be given at the first appearance.
  2. Please rewrite the abstract. I recommend consulting more literature references to use more idiomatic academic language.
  3. Why the group treated with ENR only was not set?
  4. The word of “Vibrionaceae” shouldn’t be italic.
  5. Line 66-73, the references should be added.
  6. Line 131, the word “Alosa sapidissima” should be italic.
  7. Line 136, “semi lethal concentration” or “LC50
  8. Section 3.1, the title had better revised, and we can’t find the histopathological alterations induced by ENR only, the histopathological alterations were induced by A. hydrophila+ ENR instead of ENR only.
  9. In figure 1, the first letter in horizontal and vertical coordinates should be capital, moreover, the significant difference in Figure 1A should be analyzed.
  10. Please re-photo the Figure 2, the bars in Figure 2 should be united, and high-definition image should be given.
  11. Please confirm the different letter above the column in Figure 3b.
  12. It is suggested that an expert who is fluent in English should revise the article.
  13. The Discussion section needs to be more concise.

Comments on the Quality of English Language

The present study addresses these issues to analyze the toxic effects of ENR exposure on American shad after 30 days, particularly on metabolic changes using transcriptomics and metabolomics. The article provides detailed content and clear logic. However, here are some problems that need to be appropriately addressed in the article. Particular attention should be paid to substantial improvement in writing proficiency.

  1. In Abstract, the Latin name of American shad should be given at the first appearance.
  2. Please rewrite the abstract. I recommend consulting more literature references to use more idiomatic academic language.
  3. Why the group treated with ENR only was not set?
  4. The word of “Vibrionaceae” shouldn’t be italic.
  5. Line 66-73, the references should be added.
  6. Line 131, the word “Alosa sapidissima” should be italic.
  7. Line 136, “semi lethal concentration” or “LC50
  8. Section 3.1, the title had better revised, and we can’t find the histopathological alterations induced by ENR only, the histopathological alterations were induced by A. hydrophila+ ENR instead of ENR only.
  9. In figure 1, the first letter in horizontal and vertical coordinates should be capital, moreover, the significant difference in Figure 1A should be analyzed.
  10. Please re-photo the Figure 2, the bars in Figure 2 should be united, and high-definition image should be given.
  11. Please confirm the different letter above the column in Figure 3b.
  12. It is suggested that an expert who is fluent in English should revise the article.
  13. The Discussion section needs to be more concise.

Author Response

Eidtors’ comments

Dear Dr. Zheng,

Your manuscript has been reviewed by experts in the field and we request that you make major revisions before it is processed further. Please revise your manuscript according to the reviewers' comments and upload the revised file within 10 days. Please click on the "Peer Review Reports" below to find the reviewers' comments and the version of your manuscript to be used for your revisions.

Response: Thank you for the suggestions. The authors revised the document according to the editor and reviewer’s suggestions, and the language has been polished by native English speakers.

English Language and Figures

Reviewer 2 Round 1 The English could be improved to more clearly express the research.

The present study addresses these issues to analyze the toxic effects of ENR exposure on American shad after 30 days, particularly on metabolic changes using transcriptomics and metabolomics. The article provides detailed content and clear logic. However, here are some problems that need to be appropriately addressed in the article. Particular attention should be paid to substantial improvement in writing proficiency.

Response: Thank you for the suggestions. The revised document has been polished by English native speakers, and checked again. Some sentences have been deleted to make it more clear.

In Abstract, the Latin name of American shad should be given at the first appearance.

Response: Thank you, “Alosa sapidissima A. Wilson” has been moved to line 15.

Please rewrite the abstract. I recommend consulting more literature references to use more idiomatic academic language.

Response: Thank you for the suggestions, the abstract has been revised, as “In order to find the biochemical effects of Aeromonas hydrophila and its therapeutic chemical, enrofloxacin (ENR) on American shad (Alosa sapidissima A. Wilson), four groups were set up including the control group (C), A. hydrophila group (A) , A. hydrophila+70 mg·L-1 enrofloxacin (ENR) group (E1) and A. hydrophila+140 mg·L-1 ENR group (E2), histological, enzymatic activities, transcriptome and proteomics have been performed. MDA, PPO, AKP, TNF-α, and AMPK were significantly increased, while AhR and EROD decreased in the liver of American shad after treatment with A. hydrophila. AhR and EROD showed a significant decrease in E1 group, MDA, PPO, AKP, and AMPK were significantly increased, while AhR and EROD decreased in E2 group. A. hydrophila significantly increased ferroptosis, TGF-β signaling pathway, etc., ferroptosis, pyrimidine metabolism, glycerolipid metabolism significantly increased in E1 group, while protein processing in endoplasmic reticulum significantly increased in E2 group. A total of 126 shared metabolites were found in the comparisons of A vs C, E2 vs C, and the main enriched pathway were organic oxygen compounds, lipids and lipid-like molecules. Except for fluorobenzoate degradation, the pathways of ascorbate and aldarate metabolism, pyrimidine metabolism significantly increased in A and E2 groups, which further resulted in vacuolization, cell shedding and necrosis in the liver. A. hydrophila led to a significant decrease in lipid metabolism, leading to oxidative stress and energy expenditure. The addition of ENR in aquaculture significantly enhanced liver metabolic abnormalities caused by A. hydrophila. Excessive use of ENR leads to oxidative stress in American shad, affecting its immune system as well as lipid, carbohydrate, and energy metabolism” in line 13-34.

Why the group treated with ENR only was not set?

Response: Thank you for the suggestions, the control of only ENR has not been set because ENR was used in effect with Aeromonas hydrophila. Most studies found ENR resulted in harmful effects in fish species, like the review article said “Enrofloxacin (ENR) is used to prevent and treat fish diseases widely. However, its pollution is increasing public concern on human health and aquatic ecosystem safety.”. Further studies on the effects of ENR in American shad can be performed in the future.

Xi F. The enrofloxacin pollution control from fish to the environment. Mar Pollut Bull. 2024, 199:115923. doi: 10.1016/j.marpolbul.2023.115923.

The word of “Vibrionaceae” shouldn’t be italic.

Response: Thank you, corrected as suggested in line 54.

Line 66-73, the references should be added.

Response: Thank you for the suggestions, the references [4-5] have been added in line 62.

Line 131, the word “Alosa sapidissima” should be italic.

Response: Thank you, corrected as suggested in line 130.

Line 136, “semi lethal concentration” or “LC50”

Response: Thank you, corrected as suggested in line 135.

Section 3.1, the title had better revised, and we can’t find the histopathological alterations induced by ENR only, the histopathological alterations were induced by A. hydrophila+ ENR instead of ENR only.

Response: Thank you, corrected as suggested in line 275, 286-287, 300.

In figure 1, the first letter in horizontal and vertical coordinates should be capital, moreover, the significant difference in Figure 1A should be analyzed.

Response: Thank you for the suggestions. Fig.1 changed in line 283.

Please re-photo the Figure 2, the bars in Figure 2 should be united, and high-definition image should be given.

Response: Thank you for the suggestions. Fig.2 has been changed in line 296.

Please confirm the different letter above the column in Figure 3b.

Response: Thank you for the suggestions. The letter in Fig. 3b has been changed in line 328.

It is suggested that an expert who is fluent in English should revise the article.

Response: Thank you for the suggestions. The revised document has been polished by English native speakers.

The Discussion section needs to be more concise.

Response: Thank you for the suggestions. The whole discussion has been condensed in line 578-581, 595, 608-613, 615-618.

Quality of Figures

Reviewer 2 Round 1 Figures and tables must be improved

Response: Thank you for the suggestions. Figures (1-3, 4b, 6d, 8) have been changed, while tables not changed because we have tried our best.

Reviewer#2

The present study addresses these issues to analyze the toxic effects of ENR exposure on American shad after 30 days, particularly on metabolic changes using transcriptomics and metabolomics. The article provides detailed content and clear logic. However, here are some problems that need to be appropriately addressed in the article. Particular attention should be paid to substantial improvement in writing proficiency.

 Response: Thank you for the suggestions. The whole document has been revised following the two kind reviewers’ suggestions, which has been polished by 3 native English speakers.

1.In Abstract, the Latin name of American shad should be given at the first appearance.

 Response: Thank you for the suggestions, “(Alosa sapidissima A. Wilson)” has been added in line 14.

2.Please rewrite the abstract. I recommend consulting more literature references to use more idiomatic academic language.

Response: Thank you for the suggestions, the whole abstract has been revised in line 13-34.

3.Why the group treated with ENR only was not set?

Response: Thank you for the suggestions, the usage of ENR occurred when disease happens or protection against the A. hydrophila, which existed in the pond water/sediment.

4.The word of “Vibrionaceae” shouldn’t be italic.

Response: Thank you for the suggestions, corrected as suggested in line 54.

5.Line 66-73, the references should be added.

Response: Thank you for the suggestions, the references have been added in line 62.

6.Line 131, the word “Alosa sapidissima” should be italic.

Response: Thank you for the suggestions, corrected as suggested in line 130.

7.Line 136, “semi lethal concentration” or “LC50

Response: Thank you for the suggestions, it has been changed to LC50 in line 135.

8.Section 3.1, the title had better revised, and we can’t find the histopathological alterations induced by ENR only, the histopathological alterations were induced by A. hydrophila+ ENR instead of ENR only.

Response: Thank you for the suggestions, “induced by A. hydrophila and its normal and excessive concentrations of the therapeutic drug ENR” has been added in line 286-287.

9.In figure 1, the first letter in horizontal and vertical coordinates should be capital, moreover, the significant difference in Figure 1A should be analyzed.

Response: Thank you for the suggestions, it has been changed in line 283.

10.Please re-photo the Figure 2, the bars in Figure 2 should be united, and high-definition image should be given.

Response: Thank you for the suggestions, Fig,2 has been changed in line 296.

11.Please confirm the different letter above the column in Figure 3b.

Response: Thank you for the suggestions, Fig.4b has been changed in line 328.

12.It is suggested that an expert who is fluent in English should revise the article.

Response: Thank you for the suggestions, the revised document has been polished by 3 native English speakers.

13.The Discussion section needs to be more concise.

Response: Thank you for the suggestions, the authors condensed the discussion by deleting some paragraphs, like line 578-581, 608-613, 615-618.

Round 2

Reviewer 1 Report

Comments and Suggestions for Authors

Dear authors

Your manuscript has been much approved and is now a joy to read. There are some minor things left to do, but it should be a "quick fix", and I am looking forward to seing this manuscript published

You have added some information on provinces to the text, but on L72 Lake Taihu is mentioned without stating the province, would it be possible to add that information?

L78 type-o, it should be “residues”

L118 type-o, it should be “tanks”

L150 “once per day” seems strange (multiple injections for infection not necessary) – should it instead be “…”the fish were intraperitonally injected with a single dose of A. hydrophila (LD…), after which…”

L167 “anesthezied” – change to “euthanized”. The fish cannot have been kept alive after such excessive liver sampling.

L192-193. The sentence is still strange. I suggest “The liver tissue was homogenized with PBS at a ratio of 1:9, whereafter the supernatant was collected and stored in a -20°C freezer until analysis was performed.”

L202 type-o, it should be “studies” (multiple references).

L222-231 This paragraph still has the protocol format, please amend it into correct sentences.

L269-270: If there are different significance levels there is a need to state which levels  (*, **, ***). I think what you mean is “groups with different lowercase letters have significantly different mortality”

L271: drop “its”

Figure 2: Some pictures are still a bit blurry. Is it possible to retake photos or increase sharpness in Control and E1 for 12 h, Control 24 h and E1 48 h?

L388: should it be “increased” instead of "rose”?

L450 (eg), related to Fig 6. The text still says “unchactered” which is apparently wrong as you have responded that it is “uncharted” as I suspected. Is it not possible to change (is the Figure text also automatically generated)?

L514-515. “was concerning”– I would say “is concerning”.  I think you should elaborate this a little bit and emphasize the fact that you found ENR in muscle even in the control fish, as it points to environmental contamination. A further thought – how does this affect us when we eat this fish?

L515: should be “ENR is known to…” or “ENR was shown to..”

L517: “ENR tends to”

L518: replace both “was” with “is”

L519: “explains”

L520-521: move the part about your residues to align with the sentence on L514-515 as suggested above

L522: replace “was” with “is”

L523-524: change to “exceed” and “contributes”

L543: replace “was” with “is”

L552: replace “was” with “is”

L559-575: all amendments (tracked changes) that makes the sentences appear in past tense – it should be present tense “is”, “plays” etc

The conclusion is still too long. Suggestion: remove the first three sentences (do not describe the aim or method). The next sentence is then the first (a bit amended version) “We showed that, after ENR exposure, several liver enzymes like MDA…. increased, whereas AhR and EROD significantly decreased in American shad.” The rest of the sentences are perfectly fine.

Author Response

Eidtors’ comments

Dear Dr. Zheng,

Your manuscript has been reviewed by experts in the field and we request that you make minor revisions before it is processed further. Please revise your manuscript according to the reviewers' comments and upload the revised file within 5 days. Please click on the "Peer Review Reports" below to find the reviewers' comments and the version of your manuscript to be used for your revisions.

Response: Thank you for the suggestions. The authors revised the document according to the editor and reviewer’s suggestions.

English Language and Figures

Reviewer 1 Round 2 Figures and tables can be improved

Response: Thank you for the suggestions. The Fig.2 has been revised in line 279.

Reviewer#1

Dear authors, Your manuscript has been much approved and is now a joy to read. There are some minor things left to do, but it should be a "quick fix", and I am looking forward to seing this manuscript published.

You have added some information on provinces to the text, but on L72 Lake Taihu is mentioned without stating the province, would it be possible to add that information?

Response: Thank you for the suggestions. Lake Taihu are relative with Jiangsu and Zhejiang province.

L78 type-o, it should be “residues”

Response: Thank you, corrected as suggested in line 77.

L118 type-o, it should be “tanks”

Response: Thank you, corrected as suggested in line 177.

L150 “once per day” seems strange (multiple injections for infection not necessary) – should it instead be “…”the fish were intraperitonally injected with a single dose of A. hydrophila (LD…), after which…”

Response: Thank you, corrected as suggested in line 148-9.

L167 “anesthezied” – change to “euthanized”. The fish cannot have been kept alive after such excessive liver sampling.

Response: Thank you, corrected as suggested in line 167.

L192-193. The sentence is still strange. I suggest “The liver tissue was homogenized with PBS at a ratio of 1:9, whereafter the supernatant was collected and stored in a -20°C freezer until analysis was performed.”

Response: Thank you, corrected as suggested in line 192-3.

L202 type-o, it should be “studies” (multiple references).

Response: Thank you, corrected as suggested in line 202.

L222-231 This paragraph still has the protocol format, please amend it into correct sentences.

Response: Thank you, corrected as suggested in line 222-9.

L269-270: If there are different significance levels there is a need to state which levels  (*, **, ***). I think what you mean is “groups with different lowercase letters have significantly different mortality”

Response: Thank you, corrected as suggested in line 267-8.

L271: drop “its”

Response: Thank you, corrected as suggested in line 269.

Figure 2: Some pictures are still a bit blurry. Is it possible to retake photos or increase sharpness in Control and E1 for 12 h, Control 24 h and E1 48 h?

Response: Thank you, Fig. 2 changed in line 279.

L388: should it be “increased” instead of "rose”?

Response: Thank you, corrected as suggested in line 386.

L450 (eg), related to Fig 6. The text still says “unchactered” which is apparently wrong as you have responded that it is “uncharted” as I suspected. Is it not possible to change (is the Figure text also automatically generated)?

Response: Thank you, “unchactered” was from the sequence data and changed to “un-annotated” in the figure lengend for Fig.7, the figures were automatically generated by the software.

L514-515. “was concerning”– I would say “is concerning”.  I think you should elaborate this a little bit and emphasize the fact that you found ENR in muscle even in the control fish, as it points to environmental contamination. A further thought – how does this affect us when we eat this fish?

Response: Thank you for the suggestions, “is” changed in line 512, the discussion has been elaborated in line 513-4, 524-5.

L515: should be “ENR is known to…” or “ENR was shown to..”

Response: Thank you, corrected as suggested in line 517.

L517: “ENR tends to”

Response: Thank you, corrected as suggested in line 519.

L518: replace both “was” with “is”

Response: Thank you, corrected as suggested in line 520-1.

L519: “explains”

Response: Thank you, corrected as suggested in line 514.

L520-521: move the part about your residues to align with the sentence on L514-515 as suggested above

Response: Thank you, corrected as suggested in line 514-7.

L522: replace “was” with “is”

Response: Thank you, corrected as suggested in line 521.

L523-524: change to “exceed” and “contributes”

Response: Thank you, corrected as suggested in line 523.

L543: replace “was” with “is”

Response: Thank you, corrected as suggested in line 544.

L552: replace “was” with “is”

Response: Thank you, corrected as suggested in line 553.

L559-575: all amendments (tracked changes) that makes the sentences appear in past tense – it should be present tense “is”, “plays” etc

Response: Thank you, corrected as suggested in line 562-9.

The conclusion is still too long. Suggestion: remove the first three sentences (do not describe the aim or method). The next sentence is then the first (a bit amended version) “We showed that, after ENR exposure, several liver enzymes like MDA…. increased, whereas AhR and EROD significantly decreased in American shad.” The rest of the sentences are perfectly fine.

Response: Thank you, corrected as suggested in line 579-81.

Reviewer 2 Report

Comments and Suggestions for Authors

The authors have been revised the MS following my suggestion. The MS can be accepted at the present revision.

Comments on the Quality of English Language

The authors have been revised the MS following my suggestion. The MS can be accepted at the present revision.

Author Response

Eidtors’ comments

Dear Dr. Zheng,

Your manuscript has been reviewed by experts in the field and we request that you make minor revisions before it is processed further. Please revise your manuscript according to the reviewers' comments and upload the revised file within 5 days. Please click on the "Peer Review Reports" below to find the reviewers' comments and the version of your manuscript to be used for your revisions.

Response: Thank you for the suggestions. The authors revised the document according to the editor and reviewer’s suggestions.

English Language and Figures

Reviewer 2 Round 2 Quality of English Language: The English could be improved to more clearly express the research.

Response: Thank you for the suggestions. The language, especially some past tence has been revised to present tense.

Reviewer#2

Comments and Suggestions for Authors: The authors have been revised the MS following my suggestion. The MS can be accepted at the present revision.

Comments on the Quality of English Language: The authors have been revised the MS following my suggestion. The MS can be accepted at the present revision.

Response: Thank you.